# A systematic review and meta-analysis of factors related to first line drugs refractoriness in patients with juvenile myoclonic epilepsy (JME)

Claire Fayad[1], Kely Saad[1], Georges-Junior Kahwagi[1], Souheil Hallit[2,3], Darren Griffin[4], Rony Abou-Khalil[1], Elissar El-Hayek[1] *

1 Department of Biology, Faculty of Arts and Sciences, Holy Spirit University of Kaslik-Jounieh, Lebanon, 2 School of Medicine and Medical Sciences, Holy Spirit University of Kaslik, Jounieh, Lebanon, 3 Applied Science Research Center, Applied Science Private University, Amman, Jordan, 4 School of Biosciences, University of Kent, Canterbury, United Kingdom

* elissarhayek@usek.edu.lb

**Data Availability Statement:** All relevant data are within the manuscript and its Supporting information files.

## Abstract

### Introduction

Juvenile Myoclonic Epilepsy (JME) is a prevalent form of epileptic disorder, specifically categorized within the realm of Genetic Generalized Epilepsy (GGE). Its hallmark features encompass unprovoked bilateral myoclonus and tonic-clonic seizures that manifest during adolescence. While most JME patients respond favorably to anti-seizure medication (ASM), a subset experiences refractory JME, a condition where seizures persist despite rigorous ASM treatment, often termed "Drug-Resistant Epilepsy" (DRE). This systematic review and meta-analysis aims to determine the prevalence of refractory JME, and further to identify socio-demographic, electrophysiological and clinical risk factors associated with its occurrence. Pinpointing these factors is crucial as it offers the potential to predict ASM responsiveness, enabling early interventions and tailored care strategies for patients.

### Material and methods

The systematic review and meta-analysis followed the Cochrane Handbook and adhered to the Preferred Reporting Items for Systematic Reviews and Meta-Analyses (PRISMA) guidelines. The study evaluated outcomes post ASM treatment in JME cohorts by searching papers published up to September 2023 in PubMed/MEDLINE, Scopus, and Google Scholar databases. Predefined inclusion criteria were met by 25 eligible studies, forming the basis for analysis.

### Results

A total of 22 potential risk factors for refractory JME were documented. Notably, robust risk factors for treatment resistance included Psychiatric Disorder (Odds Ratio (OR), 3.42 [2.54, 4.61] (95% Confidence Inverval (CI)), Febrile Seizures (OR, 1.83 [1.14, 2.96] (95% CI)),

**Funding:** This research project was supported by the Holy Spirit University of Kaslik - Faculty of Arts and Sciences.

**Competing interests:** The authors have declared that no competing interests exist.

Alcohol Consumption (OR, 16.86 [1.94, 146.88] (95%CI)), Aura (OR, 2.15 [1.04, 4.47] (95% CI)), childhood absence epilepsy (CAE) evolving into JME (OR, 4.54 [1.61, 12.78] (95%CI)), occurrence of three seizure types (OR, 2.96 [1.96, 4.46] (95%CI)), and Focal EEG abnormalities (OR, 1.85 [1.13, 3.01] (95%CI)). In addition, there were some non-significant risk factors for DRE because of noticeable heterogeneity.

## Conclusion

In aggregate, over 36% of JME patients demonstrated drug resistance, with seven significant risk factors closely linked to this refractoriness. The interplay between these factors and whether they denote treatment non-response or heightened disease burden remains an open question and more studies would be required to fully examine their influence.

## Introduction

Epilepsy is a neurological condition that has been observed in humans for over 5000 years. Affecting up to 1% of the population, it, unusually, shows greater incidence at the earliest and latest ends of life. It is characterized by an enduring predisposition for unprovoked epileptic seizures and by the many neurological, cognitive, and psychological consequences of this condition [1]. Seizures can be defined as a transient occurrence of signs of abnormal, excessive, or synchronous neuronal activity in a group of cerebral neurons, or in the majority of the cortex [2]. A patient must experience at least two stereotypical, unprovoked epileptic seizures to be classically diagnosed with epilepsy [3]. Some main clinical criteria include, typical age of onset, the type of seizure experienced (myoclonic, tonic-clonic, absent, tonic, atonic), the area of the brain in which the activity starts and spreads (generalized onset, focal onset, focal to bilateral tonic-clonic) and the etiology (structural brain abnormalities, infections, metabolic disorders, immune disorders and genetic causes) of the epilepsy. Indeed, any associated changes on neurological examination, can be used to determine the specific syndrome or type of epilepsy and thus, select the best treatment regime [2, 4].

Generalized seizure onset accounts for around 30–40% of patients with epilepsy, with the majority linked to a genetic predisposition; these are qualified as genetic generalized epilepsies (GGEs) [5]. Several syndromes fall under the GGE umbrella, which is the most common form of generalized epilepsy. Noticeably, GGE patients present clean brain scans and normal intellectual functioning [6]. JME, commonly referred to as "impulsive petit mal," constitutes a prevalent generalized epilepsy syndrome, encompassing 6–12% of all epilepsy cases and 25–30% of GGEs, indicating an underlying developmental disorder emerging typically around puberty and affecting diverse brain regions [7]. Primarily, bilateral and arrhythmic myoclonia affecting the upper extremities is a hallmark manifestation that characterizes the typical ictal phenomenon in JME patients, mostly occurring during awakening [8]. Generalized tonic–clonic seizures, often preceded by myoclonic jerks, prevail in over 90% of cases, while absence seizures, marked by brief duration and variable impairment of awareness, occur in approximately one-third of individuals. Electroencephalography (EEG) findings reveal a typically normal background, featuring irregular, generalized 3–5.5-Hz spike-wave and polyspike-wave activities, with a propensity to fragment during sleep. Photoparoxysmal responses, observed in 30%–90% of cases, may incite myoclonic seizures or generalized myoclonic–tonic–clonic seizures, and hyperventilation can induce generalized spike-wave discharge in a subset of

**Table 1. Antiepileptic drugs prescribed to adults diagnosed with JME [15].**

| Antiepileptic drug | Sodium Valproate | Levetiracetam | Lamotrigine | Topiramate | Zonisamide | Clobazam | Clonazepam |
|---|---|---|---|---|---|---|---|
| Evidence | Most effective clinically; Positive psychotropic effects | Less efficacious than VPA in controlling absence seizures | Synergistic effect with VPA. Could worsen MS | May be effective in GTCS | Maybe effective in MS and GTCS | Maybe effective as adjunctive | Maybe effective as adjunctive |
| Precautions | Monitor weight gain (1/3 patients); dysmetabolic syndrome | Monitor psychiatric side effects at the beginning of maintenance dose | Titrate dosage to minimize allergic risks | Observe neuropsychiatric effects | Sedation, depression, gastrointestinal problems, allergic rash | Sedation | Sedation, tolerance |

MS: Myoclonic seizires, GTCS: Generalized tonic-clonic seizure, VPA: valproate.

patients. Despite normal neuroimaging results, JME bears a genetic predisposition, potentially exhibiting a familial component. While some individuals may present with a normal developmental history, others may manifest learning disorders or ADHD [59].

About 80% of JME cases can be controlled with first line anti-seizure medication (ASM), such as sodium valproate (VPA) monotherapy, making it an epilepsy syndrome with a very good prognosis (Table 1) [9, 10]. Furthermore, a subset of patients (17%) can discontinue medication and remain seizure-free thereafter [11]. In some cases, the intensity or frequency of myoclonic episodes may diminish, rendering them less problematic as patients with JME age. For instance, it is noteworthy that relief from myoclonus is often observed after approximately 40 years in most patients. Thus, an accurate diagnosis and the use of appropriate medication can help control seizures, but there is a well-known tendency for relapse after withdrawal. In other words, the frequency of relapses in JME is the highest of all epilepsies, so most patients are forced to receive ASMs for life. Until now, however, there have been no clear indications as to when it is possible to terminate the treatment in patients on ASMs [12]. While most JME patients respond favorably to ASMs, a subset experience refractory JME, a condition where seizures persist, despite rigorous ASM treatment (often termed "Drug-Resistant Epilepsy" (DRE)). Accordingly, it is important to determine how often individuals are refractory and how commonly ASMs can be securely withdrawn to permit consistent prognostic advising [13]. Indeed, it is well documented that many JME patients show impairments to ASMs, and this can affect the development and maintenance of refractory JME [14]. These impairments are multi-factorial in origin and reflect links to various risk factors encompassing the history, pathophysiology, treatment, seizure-type, duration, psycho-social factors, onset, and severity of the disease [13]. Due to the limited number of patients and the inconsistency between studies, the precise factors that can affect the development and maintenance of this condition are not well known.

In order to address this problem and thus provide guidance for clinicians wishing to treat patients with JME, a systematic review of the available evidence is long overdue. With this in mind, the main objective of this study was to provide a broad and extensive overview of refractory JME and the prognostic risk factors associated with it. To achieve this, a multifaceted approach was implemented. First, the prevalence of refractory JME was calculated, shedding light on the scope of this challenging condition. Second, socio-demographic, clinical, and electrophysiological factors that might contribute to drug resistance in JME patients were investigated, particularly concerning their response to first-line ASMs. By conducting a comprehensive risk assessment meta-analysis based on extensive literature and data sourced from reputable databases, the aim was to identify uptodated characteristics linked to pharmacoresistance in JME patients. The overall purpose therefore was not only to contribute to a

deeper understanding of JME complexity in response to ASMs but also to raise awareness within the medical community worldwide.

## Materials and methods

Ethical committee approval was not required for this work, as it involves a systematic review without patient involvement. The systematic review and meta-analysis followed the Cochrane Handbook for Systematic Reviews- Cochrane Handbook, and adhered to the Preferred Reporting Items for Systematic Reviews and Meta-Analyses (PRISMA) guidelines [16–18]

### Data sources

A comprehensive search was conducted across multiple databases, including Scopus, PubMed/MEDLINE, and Google Scholar, without language restrictions. This search encompassed studies from their inception up to the submission date, with the primary aim of identifying published studies on risk factors for drug resistance in JME.

### Search strategy

Three reviewers collaborated to develop a search strategy with high sensitivity to gather eligible literature. To ensure a balanced approach between machine-assisted screening and human-driven systematic evidence review, we implemented a comprehensive method involving multiple reviewers, blind assessments, and a structured conflict resolution process. In the initial collection and review phase, Rayyan.ai- Rayyan.ai- was employed for efficient article screening, utilizing machine learning to prioritize results, with one reviewer conducting a detailed systematic evidence review in Sysrev—Sysrev-JME—web-based platform [19]. Two additional reviewers independently conducted blind reviews in Sysrev, ensuring impartial assessments. Conflicting assessments were resolved through Sysrev's conflict resolution feature, followed by a collaborative discussion among all three reviewers. Consensus criteria were defined during the discussion, guiding the final article selection in Sysrev. The keywords used in the search encompassed variations of "Juvenile myoclonic seizures" or "Myoclonic epilepsy," combined with terms related to risk factors, socio-demographic predictors, clinical predictors, electrophysiological predictors, and drug refractoriness or predictors of drug resistance. Further details, including the breakdown of keywords employed in each database, filtration criteria, and the resulting hit counts, can be found in S1–S3 Tables. The primary focus was to extract data related to clinical and demographic risk factors associated with poor drug outcomes in JME patients on first-line medications, including "Valproate," "Lamotrigine," "Topiramate," and "Levetiracetam" [12]; and to assess seizure refractoriness in response to ASMs. The database search was complemented with a manual search of selected article reference lists.

### Study selection

Included studies clearly reported the prevalence and risk factors associated with pharmacoresistance in JME patients treated with first-line ASMs. Thus, articles reporting seizure outcomes following ASM treatment in properly diagnosed JME patients according to international league against epilepsy (ILAE) criteria [20], were included. After eliminating duplicate records, studies found to be unrelated based on their title or abstract were excluded. Only studies that differentiated between seizure-free JME patients (control group) and seizure-resistant patients were considered. Articles in English, Turkish, Spanish, and Japanese were included, and Google Translate Google translate for documents, was used when necessary. To minimize bias,

research on pharmacological trials, observational data, articles with insufficient data or irrelevant outcomes, and single case reports were excluded. There were no restrictions on the publication period. Additionally, studies without a clear focus on drug-resistant epilepsy, concentration-controlled trials lacking a placebo-controlled group or controlled groups and adequate data were excluded. Conference abstracts, books, review articles, unpublished studies, and studies exclusively focused on epilepsy recurrence/remission without providing a definition of refractory epilepsy/DRE/pharmacoresistant epilepsy/uncontrolled epilepsy were also excluded.

Before 2010, the definition of drug-resistant epilepsy was ambiguous, with varying authors' definitions of seizure freedom and refractoriness in JME. According to the ILAE-proposed definition in 2010, medication resistance, also known as refractory JME, is characterized as the inability to achieve prolonged seizure freedom after adequate trials of two tolerated and properly selected ASM regimens (either as monotherapies or in combination) [20]. Different studies used various definitions of DRE, but the corresponding definitions for the studies included in this meta-analysis are shown in Table 2.

In this context, "drug-resistant" was defined as the presence of any seizure type despite the use of ASMs, whereas "seizure-free" was defined as the absence of any seizure types for one year according to ILAE criteria for seizure freedom [20]. Pseudo-refractory patients, who experienced seizures due to non-compliance, inadequate care, or other factors related to incorrect ASMs or lifestyle imbalances [45], were not included in the study.

## Eligibility criteria

In this study, the PICOS strategy, following the guidelines set by Santos et al. [46], was employed to assess research eligibility. The study included a diverse population, comprising children and adults aged over 10, regardless of gender, diagnosed with JME according to ILAE criteria or similar diagnostic approaches. Studies involving oral monotherapy or combinations with conventional first-line drugs like VPA as interventions were considered. Participants were categorized into two groups: Resistant and Non-Resistant, forming the control group. The analysis encompassed a wide range of outcomes, including Family History, Gender, Mean Age of Seizure Onset, Mean Age at Diagnosis, Follow-Up Time, Psychiatric Disorders, Education, Socioeconomic Status, Consanguinity, Comorbid Conditions, Alcohol Consumption, Febrile Seizures, Abnormal Neuroimaging, Clinical Phenotype, Status Epilepticus, Photosensitivity, Seizure Type, EEG Asymmetries, Focal Findings on EEG, Photoparoxysmal Response, and Aura. The studies included in the analysis enclosed various designs, including Randomized Control Trials, Quasi-Randomized Trials, and Non-Randomized Control Trials, with both blinded and non-blinded designs. These comprehensive criteria guided the selection of relevant research for the investigation.

## Data extraction

Titles and abstracts from the search results were independently reviewed, and the selection process proceeded in four steps, as illustrated in the flowchart (Fig 1). Full articles were reviewed to ensure compatibility with the inclusion criteria, following the SysRev platform-Sysrev- (JME). Tabula-tabula.technology, an open-source software, was used to extract data tables containing prognostic risk factors, drug resistance definitions, and study designs (publication year, design, size, and conflicts of interest/bias) from articles reporting clinical variables related to seizure outcomes. This process allowed data extraction in CSV format through a simple web interface running on a Java server.

**Table 2. Overview of the definitions used for the diagnosis of DRE in the included studies.**

| Author | Definition of Drug Resistance |
|---|---|
| Mor Yam (2022) [21] | Failure of adequate trials of two tolerated, appropriately chosen and used ASMs schedules |
| Asadi-Pooya (2022) [22] | Having ongoing seizures |
| Siew-Na Lim (2023) [23] | Patients who had experienced seizures in the past one year were considered to have ongoing seizures, even if the seizures occurred due to external factors such as sleep deprivation. |
| Yoshiko Hirano (2008) [24] | The treatment-resistant group consisted of those who had seizures that markedly decreased QOL for more than half a year. |
| Sarah Martin (2019) [14] | Failure of adequate trials of two tolerated, appropriately chosen and used antiepileptic drug schedules in monotherapy or combination. |
| Paola Sánchez-Zapata (2019) [25] | The failure of two adequate regimens of appropriately chosen antiepileptic drugs |
| Amy Shakeshaft (2022) [26] | Drug-resistant (either as reported or those who are not seizure-free on ≥2 ASMs) |
| Ebru AYKUTLU (2004) [27] | The occurrence of one or more generalized tonic-clonic seizures within a year or two or more myoclonic seizures within a month despite adequate monotherapy |
| Julia Höfler (2014) [28] | Not seizure-free group MS only and GTCS only persisted. |
| Kezban ASLAN (2005) [29] | Drug resistance in epilepsy refers to the inability of antiseizure medications (ASMs) to effectively control seizures in a patient. |
| Vibeke Arntsen (2017) [30] | Patients experienced ongoing seizures despite the follow-up period |
| Mirian S.B. Guaranha (2011) [31] | Unfavorable seizure control in JME patients |
| Philine Senf (2013) [32] | Ongoing occurrence of seizures in JME, aligning with the historical view of the condition as chronic. |
| Ali A. Asadi-Pooya (2014) [33] | Patients are classified based on whether they remained seizure-free during this time. |
| MARTINOVIC´ (2001) [34] | Patients with the syndrome of JME who remained uncontrolled in spite of rational AED therapy |
| FERNANDO-DONGAS (2000) [35] | Resistance was defined as recurrent seizures despite therapeutic levels (50–100 mg dl-1) of VPA. |
| Gelisse (2001) [36] | Resistant defined as persisting seizures (myoclonic jerks and/or absence seizures and/or GTCS) despite adequate lifestyle and treatment that included adequate doses of VPA |
| Manuel (2015) [37] | 'Treatment resistance' was defined as having ≥2 GTCS or disabling myoclonus resulting in falls, while on optimal dose of a first-line AEDs for JME. |
| Sager (2022) [38] | Truly resistant patients were defined as those with ongoing seizures despite recommended lifestyle and treatment with sufficient doses of VPA. |
| Jayalakshmi (2014) [39] | Lack of response to VPA in patients with JME |
| Hernández-Vanegas (2016) [40] | "Persistent seizures" were defined as the presence of any seizure type in the last year, whereas "seizure-free" was defined as a lack of any seizure types for one year according to the ILAE criteria for seizure freedom |
| Cação (2018) [41] | Refractory epilepsy, defined by the ILAE as failure of adequate trials of two tolerated and appropriately chosen and used AED schedules to attain sustained seizure freedom |
| Viswanathan (2021) [42] | Patients who had a duration of epilepsy of more than 10 years with at least 1 year of follow-up, who had complete information with respect to clinical details, seizure frequency, EEG and imaging reports and treatment history |
| Gürer (2019) [43] | Patients in whom the 2-year seizure-free period could not be achieved were included in the refractory group. |
| Chen (2020) [44] | Persistent seizures in JME, which may be related to the insensitivity of younger age to AED treatment. |

QOL: Quality of life, AED: antiepileptic drugs, ASMs: antiseizures medications, GTCS: generalized tonic-clonic seizures, MS: Myoclonic seizures, VPA: Valporate, ILAE: international league against epilepsy, EEG: electroencephalogram.

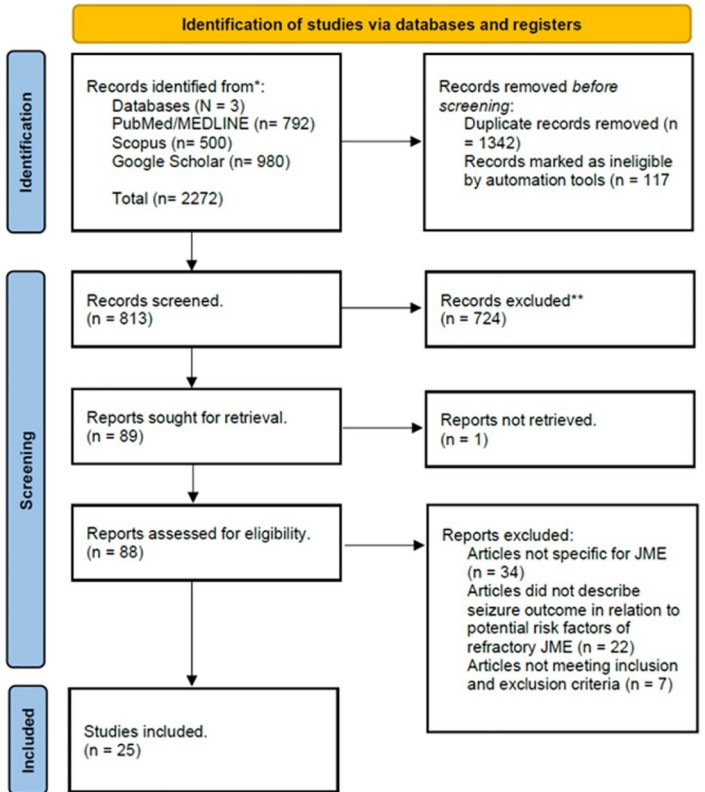

**Fig 1. The PRISMA © flowchart.** Illustration of the progression of our study, outlining the quantity of citations found in titles and abstracts, the removal of duplicates, inclusion of full texts, as well as the exclusion criteria and reasons for exclusions.

To reduce bias, raw data of potential risk factors were extracted randomly from all studies. A standardized data extraction form was created based on the assessment of the variable's association with seizure outcomes. This process was conducted by one researcher and double-checked by a second reviewer. Notably, a second independent reviewer conducted a thorough double-check of the data using a comprehensive comparison and observation approach. This meticulous multi-step verification strategy was implemented to guarantee the accuracy and reliability of the extracted data, particularly addressing potential missing data from the systems used, namely SysRev and Tabula. However, only risk factors mentioned in two or more articles were analyzed, regardless of whether they were significantly associated with the outcome.

## Bias and quality assessment

To ensure the integrity of the systematic review, a thorough evaluation of each included study was conducted, involving the independent assessment of two reviewers. The evaluation hinged on the application of the Cochrane Collaboration's widely respected Risk of Bias (ROB) assessment tool, a framework renowned for its role in evidence synthesis [47]. This tool systematically examines six key aspects of study design and execution to provide a comprehensive understanding of the strengths and limitations of each study. Firstly, it scrutinizes the adequacy of random sequence generation to ensure unbiased group allocation. Second, it assesses allocation concealment, examining whether the process of assigning participants to groups is

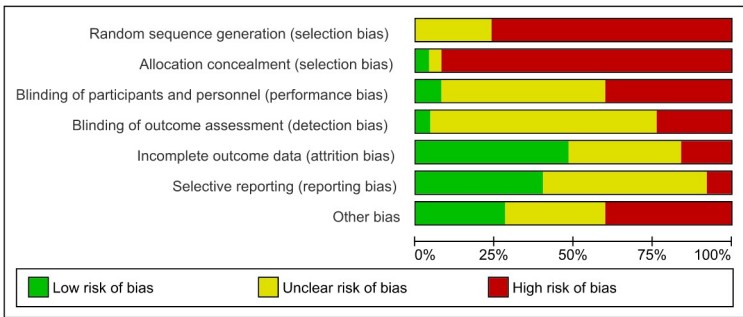

**Fig 2. Risk of bias graph.** Review of authors' judgements about each risk of bias item presented as percentages across all included studies.

transparent and unbiased. Third, the tool gauges the blinding of participants and personnel to minimize performance bias. Fourth, it evaluates the blinding of outcome assessors to prevent detection bias in outcome measurement. Fifth, it examines how incomplete outcome data, such as participant dropouts, are handled to minimize attrition bias. Lastly, it considers other potential sources of bias specific to each study, encompassing issues beyond the aforementioned factors that may impact study validity.

By systematically addressing these aspects, the ROB assessment tool provides a comprehensive evaluation of each study's methodological strengths and limitations to rate the risk of bias for each of these domains as "low risk," "high risk," or "unclear risk" based on the information provided in the study report. To quantify the judgements made by the reviewers regarding the risk of bias, we utilized RevMan software (v.5.4)- RevMan.5, a software designed for Cochrane Reviews that facilitated the presentation of authors' assessments as percentages across all the studies included in the analysis, scored on a scale ranging from 0 to 100% (Fig 2). Notably, the main focus rested on the first two bias factors—random sequence generation and allocation concealment—due to their substantial impact on our result analysis. To enhance the objectivity of the assessments, each study underwent dual evaluation by two reviewers. During this evaluation, it became evident that some included studies exhibited high risk in the first two bias factors, refer to S1 Fig.

In response to these findings, we incorporated the Newcastle–Ottawa quality assessment scale -NOS—into our methodology to provide a nuanced evaluation for these studies [48]. The detailed assessment can be accessed in the S4 Table. The Newcastle–Ottawa quality assessment scale was systematically applied to assess the methodological quality of the studies, considering three major components: cohort selection, comparability, and assessment of outcome. The scale operates on a scoring system ranging from 0 to 9, with studies considered to be of high quality if they score ≥5 and of low quality if they score <5. This additional layer of evaluation was deemed necessary to ensure a comprehensive and accurate assessment of potential biases, particularly in the context of non-randomized control studies.

## Statistical data analysis

To evaluate the prevalence of refractoriness, a random-effects meta-analysis was conducted utilizing the R package Metafor (v2.0–0)—(Metafor) [49]. The I2 statistic was employed as a measure to quantify heterogeneity, with values falling between 50% and 75% considered indicative of moderate heterogeneity, and values exceeding 75% denoting high heterogeneity. To address heterogeneity between studies, a random-effects model was applied. The assessment of

the prevalence of individuals defined as drug-resistant was similarly conducted through a random-effects meta-analysis using the Metafor package.

In addition, a meta-analysis of dichotomous (e.g., family history or gender) and continuous data (e.g., mean age of seizure onset or mean age of diagnosis) was conducted based on how data were predominantly reported in the articles. Review Manager 5.4 (RevMan.5), was utilized to assess the occurrence of drug refractoriness as an associated effect for electrophysiological, clinical and demographic risk factors in JME patients and to determine the overall percentage of drug-resistant epilepsy (DRE) in JME patients. RevMan.5, facilitated data collection, meta-analysis, and graphical presentation of results. For the analysis of dichotomous data, the results were summarized using the odds ratio (OR) estimate (with a 95% confidence interval). In the case of continuous data meta-analysis, standard mean differences were employed. Subgroup analyses were conducted as needed, focusing on variables such as gender, psychiatric disorders, and JME phenotypes. Heterogeneity between studies was evaluated by calculation of the Cochrane Q statistic [50]. Higgins I2 statistic was used to quantify the magnitude of heterogeneity, it describes the percentage of the variability and p values, classified as following: 0–40% considered as not important, 30–50% moderate, 50–75% substantial, and considerable heterogeneity for 75–100%. When $I2 > 50\%$, $Phetero < 0.1$, in other words, the heterogeneity is substantial or considerable, the random-effects model will be used. However, if moderate or not considerable heterogeneity is found ($I2 < 50\%$, $Phetero > 0.1$), the fixed-effects model was used. All possible risk factors mentioned in at least two papers were subjected to test how the study characteristics (e.g., age, sex, family history. . .) are associated with the drug refractoriness. Forest plots are employed as visual representations to illustrate the interconnection among studies and to estimate the association between drug refractoriness and the respective risk factor for each case (Fig 3), S2 File.

Statistical significance was defined as $p < 0.05$. To conduct a sensitivity analysis and ensure the robustness of the evidence synthesis, an assessment of the impact of individual studies on the pooled estimate was carried out by systematically excluding one study at a time. This approach involved removing studies one by one and examining whether the overall effect size (e.g., z-value) was significantly altered in terms of direction or magnitude. To evaluate potential publication biases, a preliminary assessment was conducted through visual inspection of funnel plots (Fig 4). However, it's important to note that a lack of symmetry was observed in some studies with small sample sizes, presented in the supplementary section S1 File.

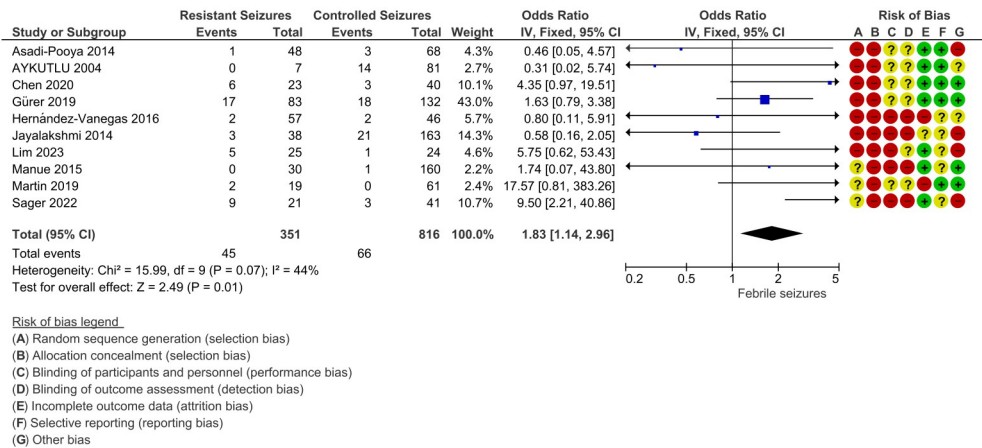

**Fig 3. Forest plot of comparison.** 1 ASM Resistant VS ASM Non-Resistant, outcome: 14 Febrile Seizures.

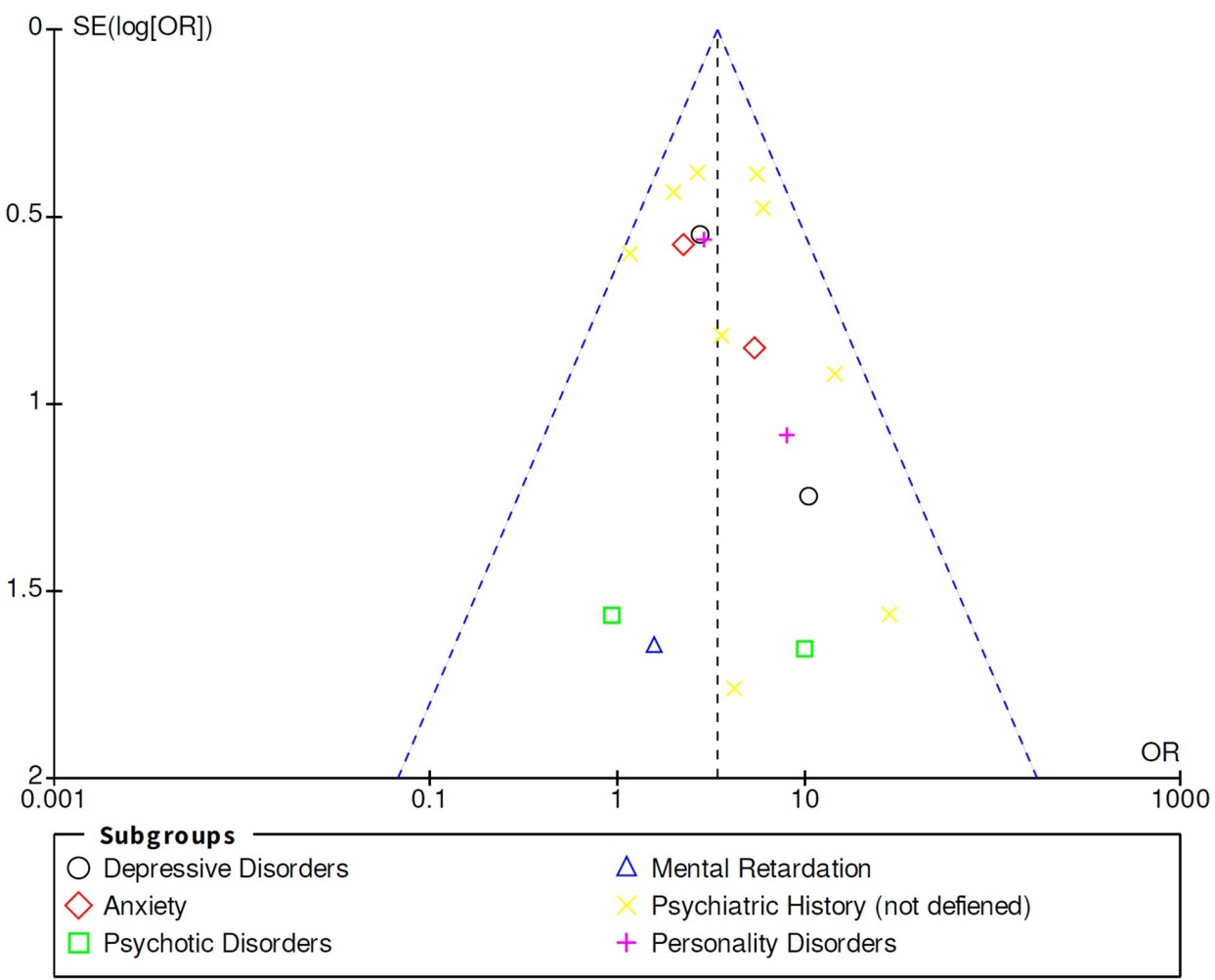

**Fig 4. Funnel plot of comparison.** 1 ASM Resistant VS ASM Non-Resistant, outcome: 5 Psychiatric disorders.

## Results

### Study search and selection

A total of 2272 records was initially identified. As stated in the flow diagram (Fig 1). 1342 duplicate records were removed, and 117 studies were marked ineligible by automation tools. Identification of 813 articles was done by title and abstract review. Irrelevant articles are excluded. In stage 2, 88 studies were reviewed in full-text study for all form eligibility. Of those 63 were excluded for not meeting the inclusion criteria, 34 were not specific for JME and included all forms of GGEs, 22 articles did not describe seizure outcome in relation to potential risk factors of refractory JME, and 7 do not comply with other inclusion and exclusion criteria. Eventually 25 studies were included in the final meta-analysis.

### Quality and characteristics of studies

The general characteristics and details of the included articles published between 2000 and 2023 are summarized in Table 2 and 3.

**Table 3. Study quality and characteristics.**

| Author | Design | Region, year | Size | Age | DRE |
|---|---|---|---|---|---|
| Mor Yam | P | Israel, 2022 | 19 | 27.27 ± 2.30 | 8 |
| Asadi-Pooya | R | Iran, 2022 | 135 | 15 (2–38) | 82 |
| Siew-Na Lim | R | Taiwan, 2023 | 49 | 27.6 ± 8.9 | 25 |
| Yoshiko Hirano | R | Tokyo, 2008 | 47 | 14 | 1 |
| Sarah Martin | R | Germany, 2019 | 87 | 8–25 | 26 |
| Paola S ánchez-Zapata | R | Colombia, 2019 | 145 | 13–16 | 51 |
| Amy Shakeshaft | R & P | London, 2022 | 765 | 23 | 165 |
| Ebru AYKUTLU | R | Istanbul, 2004 | 95 | 12.7 ±3.4 | 7 |
| Julia H öfler | R | Austria, 2014 | 175 | 15 | 66 |
| Kezban ASLAN | R & P | Adana, 2005 | 32 | 11–15 | 20 |
| Vibeke Arntsen | R | Norway, 2017 | 40 | 35–81 | 19 |
| Mirian S.B. Guaranha | R | Brazil, 2010 | 65 | 24.40 ±7.28 | 40 |
| Philine Senf | R | Germany, 2013 | 66 | 20–29 | 27 |
| Ali A. Asadi-Pooya | R | Iran, 2014 | 116 | 16 ± 3.2 | 48 |
| MARTINOVIC ´ | R | Yugoslavia, 2001 | 58 | 8 -18 | 22 |
| FERNANDO-DONGAS | R | NC, USA, 2000 | 33 | 10 -16 | 10 |
| Gelisse | R | Marseilles, Nice, 1981 and 1998 | 140 | 15–70 | 24 |
| Manuel | R | India, April 2009 to June 2011 | 44 | >12 years | 22 |
| Sager | R | Helsinki, 2020 and 2021 | 62 | 10 -18 | 21 |
| Jayalakshmi | R | India, January 2000 to January 2011 | 201 | <20 years | 38 |
| Hern ández-Vanegas | R | Mexico, 2009 to 2012 | 103 | 28.4± 7.4 | 57 |
| Ca ç ão | R | UK, 2018 | 240 | 14.2 (SD 4.5) | 121 |
| Viswanathan | R | South India (1983 –2018) | 56 | >18 years | 22 |
| G ürer | R | Turkey, 2019 | 215 | 13–16 | 83 |
| Chen | R | China, 2008 to 2013 | 63 | <16 years | 23 |

P: Prospective, R: Retrospective, DRE: Drug resistant epilepsy.

Few studies were published before the 2010 ILAE guidelines on DRE; consequently, the definition of drug resistance to first line anti-seizure medication was quite similar between all the studies. "JME patients" is the only study population characterizing the 25 studies included, conducted in Europe, Asia, and America respectively. A total of 16 cohorts included only adolescents (<18 years) and 9 cohorts grouped adult patients. The design varied between retrospective and prospective studies, whereas most study cohorts were hospital based. The sample sizes among the studies varied, ranging from 19 to 765 patients, with a total of 3,051 participants, among whom drug resistance developed in 1,028 cases.

## Prevalence of refractory Juvenile Myoclonic Epilepsy

The meta-analysis showed that 36.6% (95% (Cl), 29.8 –43.7%) of individual with JME were refractory to drugs. The proportion of resistant JME patients varied between 2.4% and 62.3%, and heterogeneity between studies is relatively high (I2 = 93%) since definitions of drug resistance slightly varied between studies specially among those before 2010 (Fig 5). Moreover, the percentage of pharmaco-resistant patients was mostly similar between prospective (21%, 42%, 62%) and retrospective studies (36%, 41%, 55%).

Then, an examination was conducted to assess whether there has been a change in the percentage of individuals who are drug resistant over time (Fig 6). Noticeably, the prevalence of

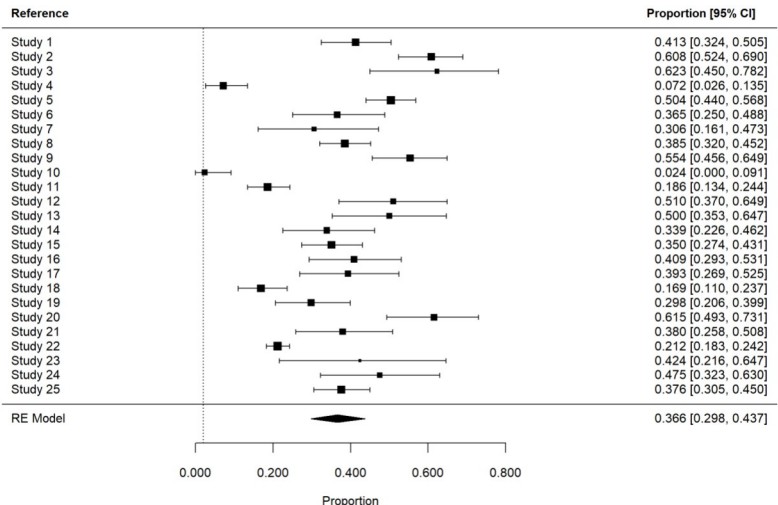

**Fig 5. Meta-analysis of the prevalence of refractory juvenile myoclonic epilepsy (JME).** The proportion of subjects who were refractory is displayed on the x-axis. A total of 25 studies describing seizure outcome in 3051 individuals with JME were included. CI, confidence interval; RE, random effects. References denoted as 'Study' are available in the S5 Table.

JME patients was slightly lower prior 2010, as it was shown for years 2004 and 2008 with 2.4% and 7.2% respectively, and this is probably for some classification reasons or medical and clinical progress, however it is relatively constant for the last 10 years.

## Risk factors of drug resistance in JME

Selected risk factors for first line medication resistance in JME patients are listed in Table 4.

The case definition of the risk factors varied across the studies, but 22 possible risk factors for refractory JME were documented in total. However, tobacco consumption and perinatal complications are 2 risk factors designated as non-applicable (NA) because they have been included in only one study. Strong risk factors for ASMs resistance were identified as Psychiatric Disorder (OR, 3.42 [2.54, 4.61] (95% Cl)), Febrile Seizures (OR, 1.83 [1.14, 2.96] (95% Cl)),

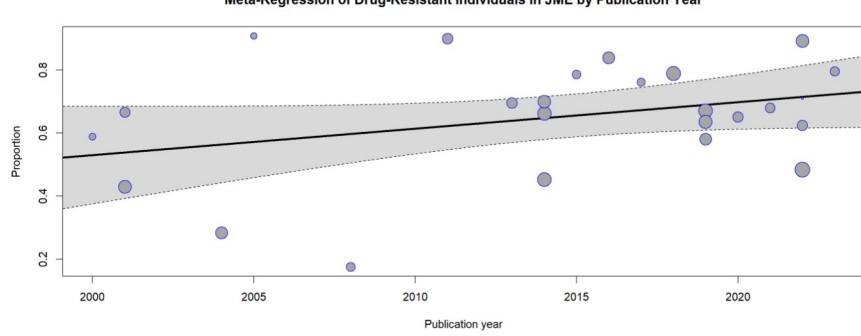

**Fig 6. Meta-regression analysis of juvenile myoclonic epilepsy refractoriness based on publication year.** We plotted the proportion of refractory subjects per study against the publication year. Each study is depicted by a circle, and the circle's size corresponds proportionally to the sample size. Additionally, a meta-regression trend line with a 95% confidence interval (represented by dotted lines) is illustrated as a solid line.

**Table 4. ASMs resistant risk factors in JME patients.**

| Risk Factors | Studies | Size | Statistical Method | Effect Estimate | I2 (%) | Phetero | P value |
|---|---|---|---|---|---|---|---|
| 1 Mean Age of Seizures Onset | 16 | 1634 | MD (IV, (F), 95% CI) | -0.35 [-0.77, 0.07] | 37 | 0.09 | 0.11 |
| 2 Mean Age at Diagnosis | 6 | 562 | MD (IV, (F), 95% CI) | -0.42 [-1.52, 0.69] | 38 | 0.19 | 0.46 |
| 3 Follow Up Time | 7 | 822 | MD (IV, (F), 95% CI) | 0.06 [-0.54, 0.67] | 12 | 0.34 | 0.84 |
| 4 Gender | 20 | 3897 | OR (IV, (F), 95% CI) | 0.97 [0.84, 1.12] | 0 | 0.57 | 0.68 |
| 4.1 Male | 16 | 1999 | OR (IV, (F), 95% CI) | 0.96 [0.79, 1.17] | 0 | 0.50 | 0.67 |
| 4.2 Female | 17 | 1898 | OR (IV, (F), 95% CI) | 0.99 [0.81, 1.20] | 0 | 0.49 | 0.88 |
| 5 Psychiatric disorders | 11 | 2346 | OR (IV, (F), 95% CI) | 3.42 [2.54, 4.61] | 0 | 0.4 | 0.00001 |
| 5.1 Depressive Disorders | 2 | 220 | OR (IV, (F), 95% CI) | 3.42 [1.28, 9.14] | 0 | 0.33 | 0.01 |
| 5.2 Anxiety | 2 | 205 | OR (IV, (F), 95% CI) | 2.96 [1.16, 7.52] | 0 | 0.4 | 0.02 |
| 5.3 Psychotic Disorders | 2 | 220 | OR (IV, (F), 95% CI) | 2.86[0.31,26.5] | 7 | 0.30 | 0.36 |
| 5.4 Mental Retardation | 1 | 140 | OR (IV, (F), 95% CI) | 1.57[0.06,39.7] | NA | NA | 0.78 |
| 5.5 Undefined Psychiatric History | 9 | 1356 | OR (IV, (F), 95% CI) | 3.52 [2.45, 5.05] | 35 | 0.14 | 0.00001 |
| 5.6 Personality Disorders | 2 | 205 | OR (IV, (F), 95% CI) | 3.58 [1.35, 9.52] | 0 | 0.40 | 0.01 |
| 6 Perinatal complications | 1 | 103 | OR (IV, (R), 95% CI) | 1.55 [0.59, 4.09] | NA | NA | NA |
| 7 Cosanguinity | 4 | 656 | OR (IV, (R), 95% CI) | 1.12 [0.56, 2.22] | 58 | 0.07 | 0.75 |
| 8 Family History | 17 | 1828 | OR (IV, (F), 95% CI) | 1.08 [0.86, 1.36] | 0 | 0.54 | 0.49 |
| 9 Tobacco consumption | 1 | 116 | OR (IV, (R), 95% CI) | 5.63 [1.12, 28.44] | NA | NA | NA |
| 10 Alcohol Consumption | 2 | 306 | OR (IV, (F), 95% CI) | 16.86 [1.94, 146.88] | 0 | 0.98 | 0.01 |
| 11 Comorbid Conditions | 5 | 693 | OR (IV, (R), 95% CI) | 3.29 [0.89, 12.10] | 79 | 0.0006 | 0.07 |
| 12 low socioeconomic status | 3 | 260 | OR (IV, (R), 95% CI) | 0.77 [0.16, 3.76] | 79 | 0.009 | 0.74 |
| 12.1 No | 2 | 115 | OR (IV, (R), 95% CI) | 0.51 [0.05, 4.77] | 87 | 0.005 | 0.55 |
| 12.2 Yes | 1 | 145 | OR (IV, (R), 95% CI) | 1.97 [0.39, 9.86] | NA | NA | 0.41 |
| 13 Education | 7 | 672 | OR (IV, (R), 95% CI) | 1.58 [0.67, 3.72] | 68 | 0.005 | 0.29 |
| 13.1 Yes | 3 | 224 | OR (IV, (R), 95% CI) | 0.79 [0.45, 1.36] | 0 | 0.61 | 0.39 |
| 13.2 No | 4 | 448 | OR (IV, (R), 95% CI) | 4.99 [0.82, 30.40] | 77 | 0.005 | 0.08 |
| 14 Febrile Seizures | 10 | 1167 | OR (IV, (F), 95% CI) | 1.83 [1.14, 2.96] | 44 | 0.07 | 0.01 |
| 15 Clinical Phenotype | 3 | 1278 | OR (IV, (R), 95% CI) | 2.27 [0.61, 8.38] | 81 | 0.0001 | 0.22 |
| 15.1 CAE evolving into JME | 2 | 455 | OR (IV, (R), 95% CI) | 4.54 [1.61, 12.78] | 0 | 0.54 | 0.004 |
| 15.2 JME with adolescent AS | 1 | 240 | OR (IV, (R), 95% CI) | 2.02 [0.49, 8.26] | NA | NA | 0.33 |
| 15.3 JME with astatic sz | 1 | 240 | OR (IV, (R), 95% CI) | 11.28 [0.62, 206.36] | NA | NA | 0.1 |
| 15.4 Classic JME | 2 | 343 | OR (IV, (R), 95% CI) | 0.82 [0.04, 15.15] | 93 | 0.0001 | 0.89 |
| 16 Seizure Type | 12 | 2874 | OR (IV, (R), 95% CI) | 1.32 [0.87, 1.99] | 66 | <0.00001 | 0.19 |
| 16.1 GTCS+AS+MJ | 10 | 973 | OR (IV, (R), 95% CI) | 2.96 [1.96, 4.46] | 19 | 0.26 | 0.00001 |
| 16.2 GTCS+MJ | 9 | 724 | OR (IV, (R), 95% CI) | 0.60 [0.28, 1.29] | 71 | 0.0009 | 0.19 |
| 16.3 AS+MJ | 4 | 398 | OR (IV, (R), 95% CI) | 2.23 [0.62, 8.09] | 0 | 0.51 | 0.22 |
| 16.4 MJ | 7 | 779 | OR (IV, (R), 95% CI) | 0.82 [0.55, 1.22] | 0 | 0.67 | 0.33 |
| 17 Abnormal Neuroimaging | 5 | 637 | OR (IV, (F), 95% CI) | 0.98 [0.66, 1.45] | 6 | 0.37 | 0.93 |
| 18 Abnormal EEG findings | 12 | 1625 | OR (IV, (R), 95% CI) | 1.98 [1.16, 3.38] | 52 | 0.008 | 0.01 |
| 18.1 EEG asymmetries | 8 | 816 | OR (IV, (R), 95% CI) | 2.16 [0.84, 5.50] | 71 | 0.0010 | 0.11 |
| 18.2 Focal Findings on EEG | 8 | 809 | OR (IV, (R), 95% CI) | 1.85 [1.13, 3.01] | 0 | 0.57 | 0.01 |
| 19 Status Epilepticus | 4 | 554 | OR (IV, (R), 95% CI) | 5.59 [0.43, 71.98] | 75 | 0.008 | 0.19 |
| 20 Photosensitivity | 5 | 670 | OR (IV, (R), 95% CI) | 0.63 [0.25, 1.58] | 56 | 0.04 | 0.32 |
| 21 Photoparoxysmal Response | 5 | 644 | OR (IV, (F), 95% CI) | 1.05 [0.63, 1.75] | 55 | 0.07 | 0.85 |
| 22 Aura | 2 | 405 | OR (IV, (F), 95% CI) | 2.15 [1.04, 4.47] | 0 | 0.5 | 0.04 |

OR: Odd Ratio, MD: Mean Difference, CI: Confidence Interval, IV: Inverse variance, CAE: Childhood absence epilepsy, GTCS: Genaralised tonic clonic seizures, AS: Absence seizures, MJ: Myoclonic jerks, F:Fixed, R:Random, NA: Non Applicable.

Alcohol Consumption (OR, 16.86 [1.94, 146.88] (95%Cl)), Aura (OR, 2.15 [1.04, 4.47] (95% Cl)), CAE evolving into JME (OR, 4.54 [1.61, 12.78] (95%CI)), occurrence of GTCS+AS+MJ (OR, 2.96 [1.96, 4.46] (95%CI)), and Focal EEG abnormalities (OR,1.85 [1.13, 3.01] (95%Cl)). Noticeably, these factors had low heterogeneity confirming their potential link to DRE. Other risk factors such as comorbid conditions (OR, 3.29 [0.89, 12.10] (95%Cl)), EEG Asymmetries (OR, 2.16 [0.84, 5.50] (95%Cl)), Low levels of education (OR, 4.99 [0.82, 30.40] (95%CI)) and status epilepticus (OR, 5.59 [0.43, 71.98] (95%CI)) were not significantly associated with DRE but this could be due to their notably high heterogeneity. Sex, family history, mean age of seizure onset and mean age of diagnosis were not significantly associated with DRE.

## Discussion

The purpose of this study was to identify demographic, clinical and electrophysiological factors associated with JME pharmaco-resistance. Notably, 36.6% of patients exhibited treatment refractoriness, consistent over a decade. Limited effective medications, like valproate, contributed to this constancy, altering the perception of JME as a benign epilepsy. Factors such as family history and drug resistance lacked significant correlation with refractoriness. However, a notable association was found between JME and psychiatric disorders, indicating a threefold increased likelihood of resistance in patients with anxiety or depression. Other factors influencing pharmaco-resistance included febrile seizures, alcohol consumption, aura, childhood absence epilepsy evolving into JME, occurrence of three seizure type, and focal EEG abnormalities. The study validates known risk factors while introducing novel insights, emphasizing the need for further epidemiological studies to address limitations in patient adherence assessment and data heterogeneity.

### DRE prevalence

A total of 36.6% of the patients that were included in the study showed treatment refractoriness. This demonstrates that the percentage of drug-refractoriness in JME remained constant over the past ten years after the establishment of a well-defined DRE characterization by the ILAE and, it is consistent with the already established prevalence of pharmaco-resistance [36]. This can be explained by the fact that there are relatively few medications that are effective against this particular form of epilepsy, such as valproate, which is still regarded as the best therapy for JME since the late 1960s [51]. These findings have led to a significant shift in the perception of JME as a benign epilepsy, necessitating caution among medical professionals when providing prognosis counseling to JME patients [23].

### Socio-demographic factors associated with DRE

Family history and drug resistance did not significantly correlate (p = 0.49, OR = 1.08) with DRE. This does not necessarily rule out a hereditary component of JME, but also it does not suggest that family history influences this condition's prognosis. There was no correlation between sex and treatment refractoriness (p = 0.68, OR = 0.97), although according to several studies, women predominate by a ratio of up to 3:1, especially between the ages of 15 and 50 [26]. This suggests that sex hormones may have a role in reducing the seizure threshold in the JME population [52]. According to Shakeshaft et al., females with JME who both have absence seizures and stress factors are three times (49%) more likely to acquire ASM resistance than their peers who do not exhibit either factor (15%) [26]. Increased risk of DRE in female could be due to valporate use restriction throughout the period of pregnancy. Accordingly, there is currently conflicting information on gender prognosis in JME. A low socioeconomic status did not predict a higher possibility of developing pharmaco-resistance (p = 0.74, OR = 0.77).

This can be explained by the improvement of the access to healthcare and the availability of generic medications across Europe, Asia, and America. Attributed to the concept of 'health literacy', recognized as the proficiency to access essential health information for making informed health decisions, low levels of education were not significantly correlated with drug-resistant epilepsy (p = 0.08, OR = 4.99). [53]. Furthermore, the age of onset was not considered a very important risk factor for DRE. Given the risk of misdiagnosis in patients under the age of 10, our meta-analysis focused on those over 10. Absence seizures, an early JME sign, can precede GTCS and myoclonic jerks by years. Without an EEG, diagnoses may be delayed, leading to misdiagnosis as childhood absence epilepsy [54]. We found controversial results in the literature as the age of seizure onset was found to be isolated from seizure outcome in two long-term studies [32, 51]; however, two other articles revealed that younger age is more likely to be associated with persistent seizures [31], raising doubts about the sensitivity of younger age to ASM therapy. There may be stability in the JME patients' response to anti-seizure medications because the likelihood of pharmaceutical resistance did not appear to relate to the delay in diagnosis (p = 0.46). A significant and well-known risk factor for JME aggravation is alcohol according to literature. Our research suggests a substantial correlation between alcohol consumption and the probability of developing pharmaco-resistance (p<0.05, OR = 16.86). Physiologically, alcohol reduces the irritability of the nervous system by turning on the GABA inhibitory pathway. As a result, EEG epileptiform activity spikes when blood alcohol levels begin to decline [56]. Active alcohol use raises blood levels of excitatory substances such as glutamate, aspartate, increasing the amount of N-methyl-D-aspartate (NMDA) subunit proteins receptors, and causes inhibition of these receptors, all this causes a rebound activation upon alcohol withdrawal [57]. Though, alcohol was proved to usually affect drugs pharmacokinetics so this may include a causality effect. Additionally, individuals with alcoholism often do not adhere to their prescribed medication, which increases the chance of developing pharmaco-resistance [58].

## Clinical factors associated with DRE

Since JME patients are not proved to acquire morphological lesions on structural brain MRIs, our meta-analysis did not find any substantial connection between the existence of neuroimaging abnormalities and refractory seizures (p = 0.93, OR = 0.98), which was an expected result [59]. Most patients were at risk of acquiring resistance to ASM if they originally displayed absence seizures in the setting of childhood absence epilepsy (CAE), followed later by a switch in their clinical condition to resemble JME (p<0.05, OR = 4.54). The electrophysiological characteristics of those two distinct GGEs syndromes, CAE and JME, and the corresponding therapy approaches vary. This suggests that CAE treatment should still be used for JME patients even if the clinical condition or EEG patterns are no longer indicating CAE (myoclonic or GTCS). Additionally, compared to those in classic JME, members of CAE evolving to JME experienced absence seizures more frequently, especially in younger ages. Consanguinity and the probability of developing drug-resistance do not significantly correlate (p = 0.75, OR = 1.12). Despite the fact that genetic epilepsies are relatively common in offspring of biologically related parents, there are not enough studies to suggest a connection between consanguinity and DRE [60]. However, JME and psychiatric disorders were significantly associated (p<0.05, OR = 3.42). Patients were almost three times more likely to experience seizure resistance if they had anxiety, depression, or other personality problems. The noradreanalin/serotonin system proposed by Jobe and Browning [61], and polyamines proposed by Baroli could explain the link between epilepsy and mental health disorders [62]. As a matter of fact, no generalizations can be drawn about the population included in our work because it is impossible

to know for sure if the patients were committed to their ASM therapy, as JME patients seems to be less adherent regularly to their medication [63]. Cognitive decline may be linked to depressive disorders, which is crucial for remembering and adhering to treatment suggestions, however, some studies suggest that there is an unclear link between ASMs regiments and psychiatric disorders seen in JME patients without a determined causality [9]. Patients with JME who had previously experienced febrile seizures were nearly twice as likely to acquire drug resistance (p<0.05, OR = 1.83). This correlation does not necessarily imply causation, but it may suggest that febrile seizures in people with a genetic predisposition to epilepsy are an early manifestation of a low seizure threshold. According to Camfield et al., persistent febrile seizures are associated with resistant epilepsy [64, 65]. Additionally, variables that increase the incidence of epilepsy following febrile convulsions may also increase the probability of unfavorable epilepsy outcomes [66]. Second, chronic febrile seizures can destroy brain tissue and cause permanent damages specially to the temporo-mesial structures, which can have negative effects on the brain including drug resistance [67].

## Electrophysiological factors associated with DRE

Status epilepticus (SE) history was not significantly associated with treatment resistance over the long term (p = 0.19, OR = 5.59). SE may have occurred due to less inhibition and hyper-excitability, and as SE sustained longer, this reduced gamma-aminobutyric acid (GABAergic) function and aggravate excitatory input [68]. Yuan et al. reported that the duration of status epilepticus when greater than 24 hours is independent predictors of DRE after status epilepticus [69]. According to further research by Oguz Akarsu et al., SE itself does not impact the outcome of JME, and the patients with SE did not experience a drug-resistant course [70]. It was discovered that having all three types of seizures—absence, myoclonic jerks, and generalized tonic-clonic seizures was a significant risk factor for developing drug resistance (p<0.05, OR = 2.96). This finding is consistent with prior long-term observational research by Hofler et al. that found individuals who experienced myoclonic jerks, absence seizures, and generalized tonic-clonic seizures within the first year of the illness were more likely to have poor seizure control [55]. When deciding whether to stop using ASMs, this clinical awareness might be viewed as a key cue. That was not applied for having other seizure types. None of the evaluated studies revealed a connection between the risk of drug resistance and patients with photosensitive JME (p = 0.32, OR = 0.63). EEG abnormality (OR, 1.98 (95%Cl)), mainly focal EEG results were discovered to be a DRE risk factor in JME (p<0.05, OR = 1.85). These asymmetric signs lead to misdiagnose JME syndrome and guide to treatment patients with carbamazepine or oxcarbazepine that were associated to aggravating effect particularly on absences and myoclonus and affect later on the sensitivity of patients to appropriate drugs [71]. Although, in some cases, patients may present focal abnormalities such as brain morphological lesions, tumors or others, not associated to the JME diagnosis cohort, but it may have a negative effect on the efficiency of ASMs. According to the ILAE, auras are a form of focal seizure, that includes autonomic, motor, psychic, sensory, or other phenomena without any disturbance of consciousness. Based on our results auras were significantly linked to JME pharmaco-resistance (p<0.05, OR = 2.15). Aura manifestation in JME varies with the brain's specific electrical activity location. For instance, visual auras are associated with the occipital lobe and may overlap with idiopathic photosensitivity or temporal lobe epilepsy. Atypical seizure characteristics including aura and post-ictal confusion were associated previously with drug resistance in JME patients [72]. Taylor et al. considered that shared genetic determinants explain why aura render both diagnosis and treatment difficult [43, 73].

## Significance of the study

Notably, Stevelink et.al. discussed the development of a predictive model for drug refractoriness in JME, which incorporates independent risk factors, such as psychiatric comorbidities, seizure types, EEG results, providing clinicians with a valuable tool for assessing the likelihood of drug resistance and informing treatment decisions in individuals with JME. Thus, our study lies in its comprehensive exploration of demographic, clinical, and electrophysiological factors influencing pharmaco-resistance in JME, it validates previously identified risk factors by stevelink et. al, while introducing additional factors associated with drug refractoriness in JME [74]. Our study contributes valuable insights for prognosis and treatment decisions. The findings challenge the perception of JME, emphasizing the need for caution in medical counseling. Moreover, the study highlights gaps in understanding, calling for further research to refine risk assessments and improve patient outcomes in JME.

## Limitation of the study

Our study has certain limitations. First of all, accurately assessing patient adherence to anti-seizure medications (ASMs) presents a significant challenge, potentially leading to misclassification of cases as "pseudo-refractory," thus affecting the accuracy of pharmaco-resistance rate estimations. Additionally, our meta-analysis faced obstacles due to the inherent data heterogeneity from various articles. Variability in drug resistance definitions, particularly in older publications, posed difficulties in synthesizing consistent findings. To overcome these limitations, additional well-designed epidemiological studies are necessary to increase the sample size, thereby ensuring more accurate and reliable results. Prioritizing the collection of data based on the ILAE definition of DRE is also needed to gain a better understanding of the correlation between DRE and its associated risk factors.

## Conclusion

In conclusion, this study highlights the complexities of managing refractory epilepsy, with a specific focus on JME. Refractory epilepsy is associated with increased morbidity, mortality, psychological issues, cognitive challenges, and a diminished quality of life. Despite the generally favorable prognosis of JME with anti-seizure medications, our meta-analysis reveals that over 36% of patients may experience drug resistance with first-line ASMs. This study identifies several key risk factors linked to this condition of drug resistance, including psychiatric disorders, alcohol consumption, focal EEG findings, aura experiences, a history of febrile seizures, childhood absence epilepsy as a clinical phenotype, and the presence of multiple seizure types. While some additional risk factors (such as education, comorbid conditions, and status epilepticus) were suggested to have a potential correlation, further research is needed due to heterogeneity and data limitations. The challenges in assessing ASM adherence and the variability in drug resistance definitions underscore the importance of more comprehensive and standardized data collection methods. To enhance our understanding of refractory JME, future studies should prioritize identifying independent predictors of drug resistance, enabling personalized predictions of seizure outcomes for tailored treatment decisions. A thorough understanding of the patient's history, clinical presentation of JME, proficiency in interpreting diagnostic tools, and neuropsychological assessment can mitigate the risk of drug resistance and enhance patients' quality of life.

Ultimately, this research contributes to the ongoing efforts of neurologists and epileptologists in diagnosing and managing JME cases effectively, aiming to improve the quality of life for individuals with this condition and potentially minimizing the adverse effects of ASMs. Thus, the identification of these factors holds significant clinical implications, equipping

neurologists with valuable insights for predicting optimal ASM responses and facilitating early-stage management of JME cases, especially those presenting risk factors associated with DRE. Conclusively, our work will, we hope, enhance the overall prospects and quality of life for individuals grappling with the challenges of JME.

## Supporting information

**S1 Fig. Risk of bias summary: Review authors' judgements about each risk of bias item for each included study.**
(TIF)

**S1 File. Funnel plot of comparison.** ASM Resistant VS ASM Non-Resistant, done for each outcome to evaluate potential publication biases.
(PDF)

**S2 File. Forest plot of comparison.** ASM Resistant VS ASM Non-Resistant, done for each outcome to depict the interconnection among studies and estimate the association between drug refractoriness and the corresponding risk factor.
(PDF)

**S1 Table. PubMed/MEDLINE search string.** The search was performed on 22 September 2023 and yielded 792 hits. Publications were filtered on the publication type "Article".
(PDF)

**S2 Table. Scopus search string.** The search was performed on 22 September 2023 and yielded 500 hits. Publications were filtered on the publication type "Article".
(PDF)

**S3 Table. Google-Scholar search string.** The search was performed on 22 September 2023 and yielded 980 hits. Publications were filtered on the publication type "Article".
(PDF)

**S4 Table. Risk of bias assessment using the Newcastle –Ottawa quality assessment scale for cohort studies.** Studies can be attributed a maximum of one star (*) for each item. The total score is calculated as the sum of stars. A higher score indicates a better quality of the study.
(PDF)

**S5 Table. Details of all 25 included studies.** A total of 25 studies describing seizure outcome in 3051 individuals with JME were classified based on the number of drug resistant individuals, number of seizure free individuals, YOS: Year of study, sample size, yi: vector with the observed effect sizes or outcomes, vi: vector with the corresponding sampling variances, pi: vector with the (signed) p-values.
(PDF)

## Author Contributions

**Conceptualization:** Claire Fayad, Georges-Junior Kahwagi, Elissar El-Hayek.

**Data curation:** Claire Fayad, Kely Saad, Elissar El-Hayek.

**Formal analysis:** Claire Fayad, Kely Saad, Elissar El-Hayek.

**Investigation:** Kely Saad, Georges-Junior Kahwagi, Elissar El-Hayek.

**Methodology:** Claire Fayad, Elissar El-Hayek.

**Supervision:** Elissar El-Hayek.

**Validation:** Elissar El-Hayek.

**Visualization:** Claire Fayad.

**Writing – original draft:** Claire Fayad, Kely Saad, Georges-Junior Kahwagi.

**Writing – review & editing:** Souheil Hallit, Darren Griffin, Rony Abou-Khalil, Elissar El-Hayek.

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
