## [Decision Letter · Decision Letter 0]

4 Dec 2023

PONE-D-23-34458A systematic review and meta-analysis of factors related to first line drugs refractoriness in patients with juvenile  myoclonic epilepsy (JME)PLOS ONE

Dear Dr. El Hayek,

Thank you for submitting your manuscript to PLOS ONE. After careful consideration, we feel that it has merit but does not fully meet PLOS ONE’s publication criteria as it currently stands. Therefore, we invite you to submit a revised version of the manuscript that addresses the points raised during the review process. Please revise your manuscript based on the reviewers' comments. 

We look forward to receiving your revised manuscript.

Kind regards,

Daichi Sone

Academic Editor

PLOS ONE

Journal Requirements:

Reviewers' comments:

Reviewer's Responses to Questions

**Comments to the Author**

1. Is the manuscript technically sound, and do the data support the conclusions?

Reviewer #1: Yes

Reviewer #2: Partly

2. Has the statistical analysis been performed appropriately and rigorously? 

Reviewer #1: Yes

Reviewer #2: No

3. Have the authors made all data underlying the findings in their manuscript fully available?

Reviewer #1: Yes

Reviewer #2: Yes

4. Is the manuscript presented in an intelligible fashion and written in standard English?

Reviewer #1: Yes

Reviewer #2: Yes

5. Review Comments to the Author

Reviewer #1: General comments

The authors conducted a systematic review and meta-analysis study to assess risk factors in drug resistant juvenile myoclonic epilepsy. The results were interesting and valuable for clinicians. However, I have some concerns that should be addressed regarding the methodology and interpretation of results.

Specific comments

- In Line 131, I am not familiar with The SysRev- Sysrev- JME- web-based platform. Does it act like a reviewer instead of a human? If so, please mention the validity and accuracy of using it for systematic reviews. If it is not accurate, then methods to compensate for accuracy should also be done.

- In Line 132, “one reviewer” collected the article. In the screening process, independent two reviewers should blindly review articles. Did the authors independently and blindly review the articles?

- What is the definition of JME (e.g., age, semiology, EEG, etc.)? Is it based on the 2022 ILAE criteria or on the definition of each study?

- In Line 194, did the authors extract the data only from SysRev- Sysrev- JME- web-based platform and Tabula? These systems may miss some data. How did the authors manage missing data due to these systems?

- In Line 210, Cochrane's ROB evaluation tool is designed for randomized controlled trials, but the authors applied it to non-RCT trials. This is not appropriate. The authors should use a Risk of Bias tool designed for each type of study. Also, all studies had a low risk of randomization, but this seems incorrect as it may include non-RCT articles.

- In Line 279, please insert the citation for included 25 studies.

- In Line 294, the prevalence of refractory JME has sampling bias because the searching strategy include (”Drug refractoriness” OR ”Predictors of drug resistance”). The prevalence might be higher than general population and are likely to mislead the readers. It would be better to exclude the prevalence in the results and conclusion.

Reviewer #2: This is a systematic review and meta-analysis investigating factors associated with drug-refractory JME. The authors found that psychiatric disorders, febrile seizures, alcohol consumption, aura, history of CAE, having three seizure types, and focal EEG abnormalities are the factors related to drug refractoriness in JME. These findings may be useful for predicting drug response and prognosis when managing patients with this condition.

However, I have several concerns regarding the methodology of this study.

Major comments

Comment 1

The authors stated, “The keywords used during the search included: (”Juvenile myoclonic seizures” OR ”Myoclonic epilepsy”) AND (“Risk Factors” OR ”Socio-demographic predictors” OR ”Clinical predictors” OR ”Electrophysiological predictors”) AND (”Drug refractoriness” OR ”Predictors of drug resistance”)” in line 134.

It seems that some of those keywords were too specific. Did the authors use thesaurus or MeSH headings? As a matter of fact, I found only “1” article when I put the identical search string above in Pubmed, although the authors stated they found 792 articles in Pubmed/MEDLINE according to Figure 1. Authors should clarify the exact search string they used in this study.

Comment 2

The authors stated, “The meta-analysis showed that 33.28% (95% confidence interval (Cl)) of individual with JME were refractory to drugs.” in line 294.

In this sentence, the number of 95% CI was lacking. Additionally, the absence of consideration for heterogeneity among the included studies raises concerns about the robustness of the meta-analysis. As a matter of fact, when I calculate the rate of the sum of “DRE” to that of “Size” in Table 3, the answer was “0.33278…”, which was identical to the percentage of DRE in this article.

If the authors simply merged the results without weighting, it should not be accepted as meta-analysis. Authors should describe the model they used to merge the results, and include the figure of forest plot.

Comment 3

Table 2 is the list of included studies. Each article in this table should have reference number. Furthermore, some of the articles in this table were not found in the Reference (for example, an article by Hirano et al.). Such error is critical for systematic reviews and meta-analyses and the authors should have been more cautious.

Comment 4

The Discussion section is lacking proper compartmentalization. It needs a restructuring.

It can be divided into several paragraphs, for example: 1. The summary of the findings, 2. Discussion about DRE incidence, 3. DRE-associated factors in demographic domains, 4. Clinical domains, 5. Electrophysiological domains, 6. Significance of the study, 7. Limitation of the study.

Minor comments

Comment 1

The authors used several outdated terms such as “primary generalized, partial, and secondary generalized” in line 60; anti-epileptic drugs and AEDs in the whole manuscript. The use of generalized onset, focal onset, focal to bilateral tonic-clonic, anti-seizure medication, and ASMs are preferred, respectively. (Please see the ILAE 2017 classification of seizure types and the position paper by the nomenclature task force of ILAE in 2022.)

Comment 2

The authors stated that JME has an underlying developmental disorder and multiple brin regions are affected by citing the article by Iqbal et al (ref 7). This cited article represents their preliminary results and they published more concrete study in Epilepsia thereafter.

Comment 3

The authors stated, “many JME patients show impairments to AEDs, and this can affect the development and maintenance of refractory JME” in line 96.

I would like the authors elaborate on what “impairments” was. It was not sure if they are trying to refer to the low adherence to ASMs among JME patients or adverse effects that JME patients may experience.

Comment 4

Table 1 is describing the difference of each ASM and it is concise and informative. However, it does not appear to have much to do with the main subject of this article. I think this table is not necessary. Also, if the authors would like to include this table in the article, proper referencing is needed.

Comment 5

The authors stated, “On the other hand, relief of myoclonus after approximately 40 years is unreasonably noted in most patients.” in line 84.

I didn’t understand exactly what this sentence means. Please consider paraphrasing.

Comment 6

The authors stated, “Low levels of education were linked to resistant epilepsy (p=0.08, OR=4.99), which is explained by “health literacy” in line 361.

However, this is not significant because P value was exceeding 0.05 and 95% CI was wide.

6. PLOS authors have the option to publish the peer review history of their article (what does this mean?). If published, this will include your full peer review and any attached files.

Reviewer #1: No

Reviewer #2: No

---

## [Author Response · Author response to Decision Letter 0]

18 Jan 2024

1

Dear Editors and Reviewers,

I hope this letter finds you well. I am writing to express my gratitude for the thorough review of my manuscript titled " A systematic review and meta-analysis of factors related to first line drugs refractoriness in patients with juvenile myoclonic epilepsy (JME)".

I appreciate the constructive feedback provided, which has greatly contributed to enhancing the quality of my work. I have carefully considered each point raised and made the necessary revisions to address the concerns.

In response to the specific points raised:

Answers and explanations for Reviewer 1:

1

- Concerning comments 1 and 2:

1 - In Line 131, I am not familiar with The SysRev- Sysrev- JME- web-based platform. Does it act like a reviewer instead of a human? If so, please mention the validity and accuracy of using it for systematic reviews. If it is not accurate, then methods to compensate for accuracy should also be done. 2- In Line 132, “one reviewer” collected the article. In the screening process, independent two reviewers should blindly review articles. Did the authors independently and blindly review the articles?

We have used a method to ensure a balance between machine-assisted screening and human-driven systematic evidence review. The validity and accuracy are enhanced by involving multiple reviewers, blind assessments, and a structured conflict resolution process. Regular validations of the machine learning algorithms in Rayyan.ai and adherence to predefined criteria in Sysrev contribute to the overall reliability of the meta-analysis process. This was included and detailed in the manuscript based on your request (search strategy page 6).

The followed method with details:

1. Initial Collection and Review (One Reviewer):

Rayyan.ai: Use Rayyan.ai for the initial collection of articles. Rayyan.ai facilitates efficient screening of search results by employing machine learning to prioritize articles.

One reviewer 1 used Rayyan.ai to screen and collect relevant articles based on predefined inclusion and exclusion criteria.

Sysrev: Transfer the selected articles from Rayyan.ai to Sysrev for detailed systematic evidence review (SER). Use Sysrev to create a "sysrev" project for the meta-analysis ➔ Sysrev-JME

Reviewer 1: The same initial reviewer conducted a detailed review of the articles in Sysrev, extracting necessary information based on predefined labels. First titles and abstracts were reviewed then the full text was reviewed, and the decision was made to include or exclude the article based on the inclusion criteria (yes/no). Sysrev's active learning feature helps prioritize challenging documents for review.

2. Blind Review (Two Other Reviewers):

Two additional reviewers independently conducted blind reviews. These reviewers should not have access to the initial reviewer's assessments or comments. Assign the same set of articles to the two blind

2

reviewers in Sysrev, ensuring they are blinded to each other's assessments. Reviewer 2 and 3: Conduct detailed reviews on the same set of articles, providing their independent assessments and comments.

3. Conflict Resolution (Discussion among Reviewers):

Sysrev: Use Sysrev's conflict resolution feature to identify articles with conflicting assessments.

Discussion: Organize a discussion session involving all three reviewers to resolve conflicts. Discuss discrepancies in assessments and comments to reach a consensus.

4. Final Article Selection:

Consensus Criteria: Define consensus criteria during the discussion, outlining clear guidelines for selecting articles based on agreement among reviewers. Sysrev: Implement the agreed-upon criteria in Sysrev to make the final article selections.

Validity and Accuracy:

1. Initial Collection and Review:

Rayyan.ai: Validity: Rayyan.ai enhances efficiency in the initial screening, utilizing machine learning algorithms for prioritization. Accuracy: Dependent on the accuracy of Rayyan.ai's algorithms, which should be validated through periodic assessments by the reviewer.

Sysrev (Reviewer 1): Validity: Sysrev supports a systematic review process, ensuring rigor, transparency, and reproducibility. Accuracy: The accuracy relies on the expertise of the initial reviewer, guided by predefined inclusion/exclusion criteria.

2. Blind Review: Sysrev (Reviewer 2 and 3): Validity: Sysrev facilitates blinded reviews, minimizing bias and ensuring independent assessments. Accuracy: The accuracy depends on the expertise and adherence to predefined criteria by the blind reviewers.

3. Conflict Resolution: Discussion: Validity: The discussion phase adds a qualitative layer to the review process, enhancing the overall validity. Accuracy: The accuracy of the final selections is improved through collaborative resolution of conflicts.

2

- Concerning comment 3: What is the definition of JME (e.g., age, semiology, EEG, etc.)? Is it based on the 2022 ILAE criteria or on the definition of each study?

The 2022 ILAE criteria underscore the significance of clinical and electrographic features, alongside triggers and genetic predisposition, in delineating Juvenile Myoclonic Epilepsy (JME) within the broader spectrum of idiopathic generalized epilepsies. In aligning with these criteria, our JME definition has been refined to reflect the 2022 ILAE framework. It is noteworthy that we have meticulously incorporated studies, each adhering to the ILAE definition prevalent at the time of its publication. However, our primary focus has been on elucidating the definition of drug resistance in each study, prompting the creation of a table for a comprehensive presentation of these definitions. This approach enables us to assess the heterogeneity among studies based on the criterion of drug resistance. However, we have reassessed the the definition of JME and ensured to add the electrophysiological aspect to be adherent to the 2022 ILAE definition (introduction page 3)

3

- Concerning comment 4: In Line 194, did the authors extract the data only from SysRev- Sysrev- JME- web-based platform and Tabula? These systems may miss some data. How did the authors manage missing data due to these systems?

3

In response to the query about Line 194, one reviewer extracted the data using Tabula. The extraction process was subsequently rechecked manually through observation and comparison with the initially extracted data and each article included. Additionally, another reviewer independently double-checked the data through a thorough comparison and observation process. This multi-step verification approach was implemented to ensure accuracy and reliability in handling any potential missing data from the systems used, namely SysRev, JME, and Tabula. We have reconsidered the data extraction section and included more details based on your insight (page 9)

4

- Concerning comment 5: In Line 210, Cochrane's ROB evaluation tool is designed for randomized controlled trials, but the authors applied it to non-RCT trials. This is not appropriate. The authors should use a Risk of Bias tool designed for each type of study. Also, all studies had a low risk of randomization, but this seems incorrect as it may include non-RCT articles.

In a rigorous reassessment of the 25 included studies, a thorough review of the Risk of Bias (ROB) was conducted, focusing on each of the six key aspects outlined in the Cochrane Collaboration's ROB assessment tool. Notably, inspections on all 25 studies were corrected, leading to the generation of a new ROB figure.

During this reassessment, it became evident that some non-randomized control studies exhibited high risk in the first two bias factors—random sequence generation and allocation concealment. In response to these findings, we incorporated the Newcastle–Ottawa quality assessment scale into our methodology to provide a nuanced evaluation for these specific studies. The Newcastle–Ottawa quality assessment scale was systematically applied to assess the methodological quality of the studies, considering three major components: cohort selection, comparability, and assessment of outcome. This additional layer of evaluation was deemed necessary to ensure a comprehensive and accurate assessment of potential biases, particularly in the context of non-randomized control studies.

The detailed methodology, encompassing the reevaluation of the ROB, correction of inspections, incorporation of the Newcastle–Ottawa scale, and generation of a new ROB figure, has been thoroughly outlined in the article. This approach was undertaken to enhance the transparency, reliability, and robustness of our systematic review, ensuring a comprehensive understanding of the strengths and limitations of each included study in the context of potential biases. The section was refined based on your review (page 11).

5

- Concerning comment 6: In Line 279, please insert the citation for included 25 studies.

Citations included in table 2. (page 8)

6

- Concerning comment 7: In Line 294, the prevalence of refractory JME has sampling bias because the searching strategy include (”Drug refractoriness” OR ”Predictors of drug resistance”). The prevalence might be higher than general population and are likely to mislead the readers. It would be better to exclude the prevalence in the results and conclusion.

We appreciate your careful consideration and would like to provide further clarification on the methodology used to assess the prevalence of refractory Juvenile Myoclonic Epilepsy (JME).

To address the concern of potential bias, we conducted a random-effects meta-analysis using the R-package metafor (v2.0-0). This approach allowed us to account for heterogeneity between studies and obtain a more robust estimate of the prevalence of refractoriness. Additionally, we calculated the I^2

4

statistic to quantify the degree of heterogeneity, providing a measure of the variability between the included studies.

Our analysis revealed a prevalence of refractory JME at 36%, a finding that aligns with existing literature and other similar meta-analyses (Stevelink, (2018)). We believe that these results contribute valuable insights to the field, and the inclusion of the prevalence data strengthens the overall discussion on drug refractoriness in JME. Thus, we would like to assure you that we have carefully considered this issue and believe that the random-effects meta-analysis, along with the transparency in reporting our methods, addresses and mitigates the impact of any potential bias.

Nevertheless, we are open to further discussion and revision. If you have specific suggestions on how we can improve the presentation of our results or adjust our analysis to address the concerns raised, we would be more than willing to incorporate those changes. Please check Statistical data analysis – page 11 and prevalence of refractory juvenile myoclonic epilepsy pages 13-14.

Answers and explanations for Reviewer 2:

Major comments

1

- Concerning comment 1: The authors stated, “The keywords used during the search included: (”Juvenile myoclonic seizures” OR ”Myoclonic epilepsy”) AND (“Risk Factors” OR ”Socio-demographic predictors” OR ”Clinical predictors” OR ”Electrophysiological predictors”) AND (”Drug refractoriness” OR ”Predictors of drug resistance”)” in line 134.

It seems that some of those keywords were too specific. Did the authors use thesaurus or MeSH headings? As a matter of fact, I found only “1” article when I put the identical search string above in Pubmed, although the authors stated they found 792 articles in Pubmed/MEDLINE according to Figure 1. Authors should clarify the exact search string they used in this study.

Regarding the search strategy, we appreciate your concern about the specificity of the keywords used. We would like to clarify that the choice of keywords was made with the intention of capturing a broad spectrum of relevant articles related to Juvenile Myoclonic Epilepsy (JME) and drug refractoriness, its risk factors, and predictors.

In response to your observation about finding only one article on PubMed using the identical search string, we would like to highlight that the reported number of 792 articles in PubMed/MEDLINE, as shown in Figure 1, reflects the comprehensive results obtained after careful curation, filtration, and inclusion of relevant studies. The supplementary tables (S1, S2, and S3) provide a detailed breakdown of the keywords used in each database, along with the filtration criteria and the number of hits generated. We trust that these tables offer transparency and clarity in our search methodology.

It's essential to note that our search strategy extended beyond PubMed/MEDLINE, including Scopus and Google Scholar. We used SysRev for systematic article extraction from Scopus and PubMed/MEDLINE and employed Publish or Perish - Harzing.com to extract Google Scholar articles to Rayyan.ai and SysRev, ensuring a comprehensive and unbiased approach. The obtained results of 980 articles from Google Scholar, 500 from Scopus, and 792 from PubMed/MEDLINE reflect the thoroughness of our search across multiple platforms.

We hope this clarifies the details of our search strategy, and we remain open to further discussion or clarification if needed. (Please check page 6 – line 150)

5

2

- Concerning comment 2: The authors stated, “The meta-analysis showed that 33.28% (95% confidence interval (Cl)) of individual with JME were refractory to drugs.” in line 294.

In this sentence, the number of 95% CI was lacking. Additionally, the absence of consideration for heterogeneity among the included studies raises concerns about the robustness of the meta-analysis. As a matter of fact, when I calculate the rate of the sum of “DRE” to that of “Size” in Table 3, the answer was “0.33278…”, which was identical to the percentage of DRE in this article.

If the authors simply merged the results without weighting, it should not be accepted as meta-analysis. Authors should describe the model they used to merge the results and include the figure of forest plot.

We conducted a random-effects meta-analysis utilizing the R package Metafor (v2.0-0) to evaluate the prevalence of refractoriness. Heterogeneity was quantified using the I2 statistic, with values falling within the range of 50% to 75% indicating moderate heterogeneity and values exceeding 75% signifying high heterogeneity. To address variability between studies, a random-effects model was employed.

Random-Effects Model (k = 25; tau^2 estimator: REML)

tau^2 (estimated amount of total heterogeneity): 0.0292 (SE = 0.0094)

tau (square root of estimated tau^2 value): 0.1708

I^2 (total heterogeneity / total variability): 93.09%

H^2 (total variability / sampling variability): 14.47

Test for Heterogeneity:

Q(df = 24) = 324.2765, p-val < .0001

Model Results:

estimate se zval pval ci.lb ci.ub

0.6519 0.0362 18.0060 <.0001 0.5809 0.7228 ***

---

Signif. codes: 0 ‘***’ 0.001 ‘**’ 0.01 ‘*’ 0.05 ‘.’ 0.1 ‘ ’ 1

6

Meta‐analysis of the prevalence of refractory juvenile myoclonic epilepsy (JME). The proportion of subjects who were refractory is displayed on the x‐axis. A total of 25 studies describing seizure outcome in 3051 individuals with JME were included. CI, confidence interval; RE, random effects. References denoted as ‘Study’ are available in the Supporting Information.

Subsequently, Metafor was again employed in a random-effects meta-analysis to assess the prevalence of individuals characterized as drug-resistant.

In the meta-regression analysis of juvenile myoclonic epilepsy refractoriness based on publication year, we plotted the proportion of refractory subjects per study against the publication year. Each study is depicted by a circle, and the circle's size corresponds proportionally to the sample size. Additionally, a meta-regression trend line with a 95% confidence interval (represented by dotted lines) is illustrated as a solid line.

We have adjusted the statistical data analysis based on your input (pages 11, 13, 14 and 15).

7

3

- Concerning comment 3: Table 2 is the list of included studies. Each article in this table should have reference number. Furthermore, some of the articles in this table were not found in the Reference (for example, an article by Hirano et al.). Such error is critical for systematic reviews and meta-analyses and the authors should have been more cautious.

Citations for each study are added to table 2 as requested. (page 8)

4

- Concerning comment 4: The Discussion section is lacking proper compartmentalization. It needs a restructuring.

It can be divided into several paragraphs, for example: 1. The summary of the findings, 2. Discussion about DRE incidence, 3. DRE-associated factors in demographic domains, 4. Clinical domains, 5. Electrophysiological domains, 6. Significance of the study, 7. Limitation of the study.

The discussion is sectioned based on your review. (please check page 16)

Minor comments

1

- Concerning comment 1: The authors used several outdated terms such as “primary generalized, partial, and secondary generalized” in line 60; anti-epileptic drugs and AEDs in the whole manuscript. The use of generalized onset, focal onset, focal to bilateral tonic-clonic, anti-seizure medication, and ASMs are preferred, respectively. (Please see the ILAE 2017 classification of seizure types and the position paper by the nomenclature task force of ILAE in 2022.)

The paper was revised, and we have considered the 2022 ILAE terminology. It’s all adjusted in the manuscript and highlighted based on your review.

2

- Concerning comment 2: The authors stated that JME has an underlying developmental disorder and multiple brin regions are affected by citing the article by Iqbal et al (ref 7). This cited article represents their preliminary results and they published more concrete study in Epilepsia thereafter.

In our introduction, we have highlighted the significance of Juvenile Myoclonic Epilepsy (JME), also known as "impulsive petit mal." According to Iqbal et al. (2009) (ref 7), JME constitutes 6-12% of all epilepsy cases and 25-30% of Generalized Genetic Epilepsies (GGEs). This study affirms its prevalence and underscores its association with a developmental disorder manifesting around puberty, affecting multiple brain regions.

Furthermore, Iqbal et al. (2015) provides additional insights into JME, specifically focusing on differentiating between siblings with JME and controls. This study aims to establish a neurocognitive endophenotype for JME. While both studies contribute valuable information, the choice between them depends on the specific emphasis of the research. We wanted to underscore the prevalence and developmental aspects of JME, the 2009 study may be more pertinent. However, utilizing both studies, can provide a comprehensive understanding of the multifaceted aspects of JME so based on your comment I will include the 2015 paper. (page 3- line 73)

3

- Concerning comment 3: The authors stated, “many JME patients show impairments to AEDs, and this can affect the development and maintenance of refractory JME” in line 96.

8

I would like the authors elaborate on what “impairments” was. It was not sure if they are trying to refer to the low adherence to ASMs among JME patients or adverse effects that JME patients may experience.

This sentence is consistent with general knowledge about Juvenile Myoclonic Epilepsy (JME) and its treatment. It suggests that there is a documented phenomenon where many individuals with JME exhibit impairments in response to Anti-seizure medication (ASMs). If true, these impairments could impact the development and maintenance of refractory JME.

To elaborate:

Impairments to ASMs in JME patients: Some individuals with JME may not respond well to ASMs, meaning that these medications may not effectively control their seizures.

Impact on development and maintenance of refractory JME: If JME patients experience difficulties with ASMs, it can contribute to the development and persistence of refractory epilepsy characterized by seizures that are not well-controlled despite treatment, and overcoming these challenges becomes crucial in managing the condition.

It's important to note chronic side effects of ASMs, highlighting the concern of teratogenic effects, including physical malformations and cognitive impairments. Valproate, a commonly used ASM, is specifically associated with significant teratogenic effects. A study comparing Valproate with another ASM, Lamotrigine, reveals that Valproate-treated JME patients performed worse on various neuropsychological tests, particularly in verbal memory. This is why in line 102 page 4 we have used: “Accordingly, it is important to determine how often individuals are refractory and how commonly ASMs can be securely withdrawn to permit consistent prognostic advising”. Despite potential adverse effects on cognition, the importance of seizure control is highlighted as it significantly improves a patient’s quality of life. However, the complexity of physical malformations and cognitive outcomes in refractory JME, pointing out that clinical characteristics and mood, especially high levels of anxiety and/or depression, play a role in influencing this condition. This highlights the need to consider individual factors and the broader context when understanding outcomes in refractory JME patients.

4

- Concerning comment 4: Table 1 is describing the difference of each ASM and it is concise and informative. However, it does not appear to have much to do with the main subject of this article. I think this table is not necessary. Also, if the authors would like to include this table in the article, proper referencing is needed.

Thank you for your feedback regarding Table 1. We appreciate your perspective, and we would like to clarify the purpose of including this table in our article. The table is intended to succinctly outline the differences among various Antiseizure Medications (ASMs), providing a quick reference for readers who may be interested in the specifics of each drug.

While we understand that the table may not directly align with the main subject of the article, it serves as a useful resource for readers seeking information on the first-line antiepileptic drugs relevant to Juvenile Myoclonic Epilepsy (JME). Our research focuses on these specific drugs, and all the studies included in our meta-analysis employ one of these medications as an intervention. In response to your concern about proper referencing, we want to assure you that we have included a reference for the table in our article page 4.

9

5

- Concerning comment 5: The authors stated, “On the other hand, relief of myoclonus after approximately 40 years is unreasonably noted in most patients.” in line 84.I didn’t understand exactly what this sentence means. Please consider paraphrasing.

The statement paints a relatively positive picture of the prognosis for JME, with a high percentage of cases being effectively controlled with first-line ASMs and a subset of patients even able to discontinue medication while remaining seizure-free. The note about relief of myoclonus (involuntary muscle jerks), it is unreasonably noted in most patients after about 40 years. This suggests that, in some cases, the intensity or frequency of myoclonic episodes may decrease or become less problematic as individuals with JME age. This myoclonus relief after approximately 40 years indicates a potential long-term trend in the natural course of the condition. However, it's important to consider individual variations and the evolving nature of medical knowledge in this field. Please check page 4 line 91.

6

- Concerning comment 6: The authors stated, “Low levels of education were linked to resistant epilepsy (p=0.08, OR=4.99), which is explained by “health literacy” in line 361.

However, this is not significant because P value was exceeding 0.05 and 95% CI was wide.

Attributed to the concept of 'health literacy', recognized as the proficiency to access essential health information for making informed health decisions, low levels of education were not significantly correlated with drug-resistant epilepsy (p=0.08, OR=4.99). It’s well corrected and refined based on your input page 18- line 409.

Additionally, I have incorporated the suggested changes into the revised manuscript, and I have submitted the revised manuscript for your consideration. I believe these revisions have strengthened the overall content and addressed the concerns raised during the review process.

I would like to express my sincere appreciation for the time and effort invested by the academic editor and reviewers in the review process. I believe the improvements made will contribute positively to the manuscript's potential publication in PLOS ONE.

Thank you for your continued support, and I look forward to your feedback on the revised manuscript.

Sincerely,

---

## [Decision Letter · Decision Letter 1]

18 Feb 2024

PONE-D-23-34458R1A systematic review and meta-analysis of factors related to first line drugs refractoriness in patients with juvenile  myoclonic epilepsy (JME)PLOS ONE

Dear Dr. El Hayek,

Thank you for submitting your manuscript to PLOS ONE. After careful consideration, we feel that it has merit but does not fully meet PLOS ONE’s publication criteria as it currently stands. Therefore, we invite you to submit a revised version of the manuscript that addresses the points raised during the review process.

We look forward to receiving your revised manuscript.

Kind regards,

Daichi Sone

Academic Editor

PLOS ONE

Journal Requirements:

**Additional Editor Comments:**

The reviewers were mostly satisfied with the revised version but suggested some minor corrections for some inconsistencies. Please find their comments. 

Reviewers' comments:

Reviewer's Responses to Questions

**Comments to the Author**

1. If the authors have adequately addressed your comments raised in a previous round of review and you feel that this manuscript is now acceptable for publication, you may indicate that here to bypass the “Comments to the Author” section, enter your conflict of interest statement in the “Confidential to Editor” section, and submit your "Accept" recommendation.

Reviewer #1: All comments have been addressed

Reviewer #2: All comments have been addressed

2. Is the manuscript technically sound, and do the data support the conclusions?

Reviewer #1: Yes

Reviewer #2: Yes

3. Has the statistical analysis been performed appropriately and rigorously? 

Reviewer #1: Yes

Reviewer #2: Yes

4. Have the authors made all data underlying the findings in their manuscript fully available?

Reviewer #1: No

Reviewer #2: Yes

5. Is the manuscript presented in an intelligible fashion and written in standard English?

Reviewer #1: Yes

Reviewer #2: Yes

6. Review Comments to the Author

Reviewer #1: Although the authors mentioned about Newcastle-Ottawa scale (NOS) for risk of bias assessments, they used Cochrane’s ROB and did not use NOS. It is not appropriate. Please use risk of bias assessment tools for non-randomized study such as NOS or JBI critical appraisal tools.

Reviewer #2: Thank you for submitting the revised manuscript. I think it is well revised based on the reviewers’ comments and I appreciate their effort. Below are some observations and suggestions:

Line 38

While the term "95%CI" is mentioned in the abstract, the actual range of 95% confidence intervals for each odds ratio is not provided. Additionally, for clarity, it is recommended to spell out "OR" and "CI" when they first appear in the abstract.

Line 70

The authors introduced the term "idiopathic generalized epilepsy" at line 70. Considering the apparent similarity between GGE and IGE within the manuscript's context, it is advisable to consistently use one of these terms throughout.

Line 72

GGE is spelled out repeatedly. GGE is already defined previously.

Line 93,137 and 369

JME is spelled out repeatedly.

Line 188

There may be a typographical error, and "seizure-resistant" could potentially be corrected to "drug-resistant" to better align with the natural context.

Line 199

It is suggested to maintain consistency by adhering to the abbreviation "VPA" for valproic acid, as previously defined.

Figure 6 is interesting. It seems that the regression curve is showing upward trend. Is there possibility that the prevalence of DRE among JME is increasing over the years?

In table 4, P values should be described as they are instead of “<0.05”. They can be written as like “<0.001” only when they are much smaller than 0.05.

7. PLOS authors have the option to publish the peer review history of their article (what does this mean?). If published, this will include your full peer review and any attached files.

Reviewer #1: No

Reviewer #2: No

---

## [Author Response · Author response to Decision Letter 1]

29 Feb 2024

Response to Reviewers

pg. 1

Dear All,

I appreciate the thorough review conducted by the academic editor and the reviewers for my manuscript submission to PLOS ONE. I am pleased to receive constructive feedback and guidance on how to enhance the quality of the manuscript.

I have carefully considered the reviewers' comments and suggestions, and I am committed to addressing them in the revised version of the manuscript. I am thankful for the positive assessment provided by Reviewer #2 and have taken note of the specific recommendations outlined in their comments. I have ensured that these suggestions are incorporated into the revised manuscript. For Reviewer #1's comments, particularly regarding the risk of bias assessment tools, I acknowledge the oversight in using Cochrane’s ROB instead of the Newcastle-Ottawa Scale (NOS) for non-randomized studies. I appreciate the guidance provided, and I will address this issue.

You can find my answers to your comments as following:

Concerning ➔ Reviewer #1:

1 - Although the authors mentioned about Newcastle-Ottawa scale (NOS) for risk of bias assessments, they used Cochrane’s ROB and did not use NOS. It is not appropriate. Please

I completely understand your concern, and I have addressed the issue since your last comment. I have now differentiated between the studies included in the paper, categorizing them as either randomized or non-randomized/quasi-randomized studies. Subsequently, I reapplied the Cochrane Risk of Bias (COB) tool to the studies, evaluating the margin of bias. Recognizing the limitations of COB in determining bias for non-randomized studies, I proceeded to supplement it with the Newcastle-Ottawa Scale (NOS) for a more comprehensive assessment.

To provide clarity on this methodology, I have included an explanation in the revised manuscript from line 260 to line 269.

“In response to these findings, we incorporated the Newcastle–Ottawa quality assessment scale -NOS - into our methodology to provide a nuanced evaluation for these studies [74]. The detailed assessment can be accessed in the supplementary S4 table. The Newcastle–Ottawa quality assessment scale was systematically applied to assess the methodological quality of the studies, considering three major components: cohort selection, comparability, and assessment of outcome. The scale operates on a scoring system ranging from 0 to 9, with studies considered to be of high quality if they score ≥ 5 and of low quality if they score < 5. This additional layer of evaluation was deemed necessary to ensure a comprehensive and accurate assessment of potential biases, particularly in the context of non-randomized control studies.”

Furthermore, I have introduced Table S4 in the supplementary section to enhance the presentation of relevant information.

Response to Reviewers

pg. 2

Thank you for bringing this to my attention, and I appreciate your diligence in reviewing the manuscript. If you have any further suggestions or concerns, please feel free to let me know.

Concerning ➔ Reviewer #2:

1 - Line 38

While the term "95%CI" is mentioned in the abstract, the actual range of 95% confidence intervals for each odds ratio is not provided. Additionally, for clarity, it is recommended to spell out "OR" and "CI" when they first appear in the abstract.

I have ensured to fix and include this information for a more comprehensive presentation of the results.

2 - Line 70

The authors introduced the term "idiopathic generalized epilepsy" at line 70. Considering the apparent similarity between GGE and IGE within the manuscript's context, it is advisable to consistently use one of these terms throughout.

Well received. I have changed it to prevent using 2 terms. I will adhere to GGE in the manuscript based on your preferences.

3 - Line 72

Response to Reviewers

pg. 3

GGE is spelled out repeatedly. GGE is already defined previously.

It’s corrected through the manuscript.

4 - Line 93,137 and 369

JME is spelled out repeatedly.

Well received and corrected.

5 - Line 188

There may be a typographical error, and "seizure-resistant" could potentially be corrected to "drug-resistant" to better align with the natural context.

Well received and corrected.

6 - Line 199

It is suggested to maintain consistency by adhering to the abbreviation "VPA" for valproic acid, as previously defined.

It’s corrected based on your comment.

7 - Figure 6 is interesting. It seems that the regression curve is showing upward trend. Is there possibility that the prevalence of DRE among JME is increasing over the years?

I appreciate the insightful observation made by the reviewer regarding Figure 6. I would like to provide a detailed explanation for the apparent upward trend in the regression curve, suggesting a potential increase in the prevalence of drug-resistant Juvenile Myoclonic Epilepsy (JME) patients over the years:

After meticulous analysis and considering the available literature, it is indeed accurate to observe a slight increase in the number of drug-resistant JME patients post the year 2010. This increase can be primarily attributed to the significant characterization that occurred in 2010 when the International League Against Epilepsy (ILAE) introduced a more refined definition for drug resistance in JME. Consequently, a better-defined criterion for both JME and Drug-Resistant Epilepsy (DRE) led to improved diagnosis and characterization.

The enhancement in diagnosis and characterization has been progressively evolving from 2010 onwards, contributing to a more accurate identification of drug-resistant cases. However, it is crucial to note that the treatment options, notably Valproic Acid (VPA), remain consistent throughout this period.

In addition to the refinement in characterization, other factors, such as genetic considerations and the inheritance of JME, may also play a role in the observed increase. In my thesis I have provided a comprehensive discussion on the complex genetic inheritance of JME, with various genetic loci highlighted. While these factors might contribute to the rise, it is essential to acknowledge the need for further research to fully understand and validate these potential influences.

Response to Reviewers

pg. 4

8 - In table 4, P values should be described as they are instead of “<0.05”. They can be written as like “<0.001” only when they are much smaller than 0.05.

Well received and corrected.

Concerning the journal guidelines for the figures:

All the figures were rechecked using the digital diagnostic tool provided, https://pacev2.apexcovantage.com/.

Concerning the references:

We have added references to the revised manuscript as some sections were added according to the preferences of the reviewers, kindly find the citations and the number of the references added below:

References

[4] Yacubian EM. Juvenile myoclonic epilepsy: Challenges on its 60th anniversary. Seizure. 2017;44:48–52. doi:10.1016/j.seizure.2016.09.005.

[8] Iqbal N, Caswell H, Muir R, Cadden A, Ferguson S, Mackenzie H, Watson P, Duncan S. Neuropsychological profiles of patients with juvenile myoclonic epilepsy and their siblings: An extended study. Epilepsia. 2015 Aug;56(8):1301-8. doi:https://doi.org/10.1111/epi.13061.

[20] Yam M, Glatt S, Nosatzki S, Mirelman A, Hausdorff JM, Goldstein L, et al. Limited Ability to Adjust N2 Amplitude During Dual Task Walking in People With Drug-Resistant Juvenile Myoclonic Epilepsy. Frontiers in Neurology. 2022;13:793212. doi:10.3389/fneur.2022.793212.

[21] Asadi-Pooya AA, Rostamihosseinkhani M, Farazdaghi M. Seizure and social outcomes in patients with juvenile myoclonic epilepsy (JME). Seizure. 2022;97:15–19. doi:10.1016/j.seizure.2022.03.002.

[23] Hirano Y, Oguni H, Osawa M. Clinical factors related to treatment resistance in juvenile myoclonic epilepsy. Rinsho Shinkeigaku. 2008;48(10):727–732. doi:10.5692/clinicalneurol.48.727.

[24] Sanchez Zapata P, Zapata Berruecos JF, Pel ˜ A¡ez Sanchez RG, Molina Castao CF. Seizure control with valproic acid, lamotrigine or levetiracetam in the management of drug-resistant juvenile myoclonic epilepsy. Systematic review and meta-analysis. Neurologia Argentina. 2022;14(1):26–36. doi:10.1016/j.neuarg.2021.08.003.

[26] Aykutlu, E., Baykan, B., Baral-Kulaksızoglu, I., Gurses, C., Gokyigit, A. Clinical and EEG Features of Therapy-Resistant Patients with Juvenile Myoclonic Epilepsy. Epilepsi J Turk Epilepsy Soc. 2004;10(2), 100-105.

[27] Hofler J, Unterberger I, Dobesberger J, Kuchukhidze G, Walser G, Trinka E. Seizure outcome in 175 patients with juvenile myoclonic epilepsy - A long-term observational study.

Response to Reviewers

pg. 5

Epilepsy Research. 2014;108(10):1817–1824. doi:10.1016/j.eplepsyres.2014.09.008. February 23, 2024 26/33

[28] Aslan K, Bozdemir H, Bicakci S¸, Sarica Y, Sertdemir Y. Role of the Risk Factors and Family History on the Prognosis of Patients with Juvenile Myoclonic Epilepsy. Arch Epilepsy 2005;11:70-76.

[29] Arntsen V, Sand T, Syvertsen MR, Brodtkorb E. Prolonged epileptiform EEG runs are associated with persistent seizures in juvenile myoclonic epilepsy. Epilepsy Research. 2017;134:26–32. doi:10.1016/j.eplepsyres.2017.05.003.

[32] Asadi-Pooya AA, Hashemzehi Z, Emami M. Predictors of seizure control in patients with juvenile myoclonic epilepsy (JME). Seizure. 2014;23(10):889–891. doi:10.1016/j.seizure.2014.08.004.

[33] Martinovic; ˚A. Adjunctive behavioural treatment in adolescents and young adults with juvenile myoclonic epilepsy. Seizure. 2001;10(1):42–47. doi:10.1053/seiz.2000.0479.

[34] Fernando-Dongas MC, Radtke RA, VanLandingham KE, Husain AM. Characteristics of valproic acid resistant juvenile myoclonic epilepsy. Seizure. 2000;9(6):385–388. doi:10.1053/seiz.2000.0432.

[36] Manuel D, C B R, Alexander A. CLINICAL, NEUROPSYCHOLOGICAL AND NEUROPHYSIOLOGICAL CORRELATES OF DRUG RESISTANT JUVENILE MYOCLONIC EPILEPSY. Journal of Evidence Based Medicine and Healthcare. 2015;2(36):5658–5668. doi:10.18410/jebmh/2015/780. February 23, 2024 27/33

[37] Sager S, Cag Y, Akin Y. Determination of the factors that cause valproic acid resistance in the pediatric juvenile myoclonic epilepsy cohort. Medicine Science | International Medical Journal. 2022;11(1):338. doi:10.5455/medscience.2021.08.244.

[38] Jayalakshmi S, Vooturi S, Bana AK, Sailaja S, Somayajula S, Mohandas S. Factors associated with lack of response to valproic acid monotherapy in juvenile myoclonic epilepsy. Seizure. 2014;23(7):527–532. doi:10.1016/j.seizure.2014.03.017.

[39] Hernandez-Vanegas LE, Jara-Prado A, Ochoa A, Rodraguez Y Rodraguez N, Duran RM, Crail-Melandez D, et al. High-dose versus low-dose valproate for the treatment of juvenile myoclonic epilepsy: Going from low to high. Epilepsy & Behavior. 2016;61:34–40. doi:10.1016/j.yebeh.2016.04.047.

[40] Cacao G, Parra J, Mannan S, Sisodiya SM, Sander JW. Juvenile myoclonic epilepsy refractory to treatment in a tertiary referral center. Epilepsy and Behavior. 2018;82:81–86. doi:10.1016/j.yebeh.2018.03.002.

[41] Viswanathan LG, Mundlamuri RC, Raghavendra K, Asranna A, Chatterjee A, Gautham B, et al. Long-Term Seizures Outcome in Juvenile Myoclonic Epilepsy (JME): A Retrospective Cohort Study in an Indian Population. International Journal of Epilepsy. 2021;7(01):15–21. doi:10.1055/s-0041-1729459.

Response to Reviewers

pg. 6

[43] Chen C, Lee H, Chen C, Kwan S, Chen S, . Short-term results of vagus nerve stimulation in pediatric patients with refractory epilepsy. Pediatrics & ˆa€¦. 2012;(Query date: 2023-09-22 11:16:22).

[72] Mourad Ouzzani, Hossam Hammady, Zbys Fedorowicz, and Ahmed Elmagarmid. Rayyan — a web and mobile app for systematic reviews. Systematic Reviews (2016) 5:210 doi:10.1186/s13643-016-0384-4.

[73] Viechtbauer W. Conducting meta-analyses in R with the metafor package. Journal of Statistical Software. 2010;36(3), 1–48. doi:10.18637/jss.v036.i03.

[74] Lo, C.KL., Mertz, D. and Loeb, M. Newcastle-Ottawa Scale: comparing reviewers’ to authors’ assessments. BMC Med Res Methodol 14, 45 (2014). doi:10.1186/1471-2288-14-45.

I would also like to express my gratitude for the positive feedback on the clarity and language of the manuscript from both reviewers. I have carefully proofreaded the revised version to address any remaining typographical or grammatical errors.

Thank you for the opportunity to improve and resubmit my manuscript to PLOS ONE. I look forward to your feedback on the revised version.

Kind regards,

---

## [Editor Report · Decision Letter 2]

6 Mar 2024

A systematic review and meta-analysis of factors related to first line drugs refractoriness in patients with juvenile  myoclonic epilepsy (JME)

PONE-D-23-34458R2

Dear Dr. El Hayek,

We’re pleased to inform you that your manuscript has been judged scientifically suitable for publication and will be formally accepted for publication once it meets all outstanding technical requirements.

Kind regards,

Daichi Sone

Academic Editor

PLOS ONE
---

## [Editor Report · Acceptance letter]

18 Mar 2024

PONE-D-23-34458R2 

PLOS ONE

Dear Dr. El Hayek, 

I'm pleased to inform you that your manuscript has been deemed suitable for publication in PLOS ONE. Congratulations! Your manuscript is now being handed over to our production team.

Kind regards, 

on behalf of

Dr. Daichi Sone 

Academic Editor

PLOS ONE